# Evolution of linear triterpenoid biosynthesis within the *Euphorbia* genus

Tomasz Czechowski[1], Yi Li[1], Alison D. Gilday[1], David Harvey[1], Sandesh H. Swamidatta [1], Benjamin R. Lichman [1], Jane L. Ward [2] & Ian A. Graham [1] ✉

Terpenoids are among the largest classes of plant natural products. Squalene, a high value commodity in the cosmetic, food and pharmaceutical industries, is a common linear precursor for the biosynthesis of C30 triterpenes and sterols across plant, animal and fungal kingdoms. The anti-fungal compound peplusol is another linear C30 triterpene, but has only been reported in the genus *Euphorbia*. Here, we show that peplusol production has evolved due to duplication of a sterol synthase gene with one copy acquiring peplusol synthase activity and the original gene retaining the ancestral function. We identify a number of key amino acid residues that can convert the squalene synthase enzyme into peplusol synthase and vice versa. The *PEPLUSOL SYNTHASE* gene from *E. peplus* is able to drive significant levels of peplusol production in alternate host production platforms including *Nicotiana benthamiana* (over 2.5% leaf dry weight) and *Saccharomyces cerevisiae* (30 mg/L culture).

Triterpenes are a class of organic compounds characterised by a structural backbone of six isoprene units resulting in a 30-carbon framework. Found in animals, plants, fungi and some bacteria, triterpenes play pivotal roles in various biological processes. In eukaryotes, sterols are synthesised via the mevalonate pathway with 2,3-oxidosqualene being the last common intermediate[1]. Sterols are triterpenes with four fused rings arranged in a specific molecular configuration that are important structural components of membranes and also have roles in signalling (as steroidal hormones)[2]. In plants, triterpenes have various other roles including as components of surface waxes and as complex glycosylated saponins which provide protection against pathogens and pests[3]. Due to their functionality simple and complex triterpenes have been developed for a wide range of applications in the food, health, and industrial biotechnology sectors[4–6].

*Euphorbia* (spurge) is one of the largest genera of flowering plants with over 2000 species. Molecular phylogenetic studies of *Euphorbia* identified four subgenera: Esula, Athymalus, Chamaesyce and Euphorbia[7,8]. Genome assemblies of representative species of the *Euphorbia* genus have been reported, such as the Esula subgenus

species *E. peplus*[9] and *E. lathyris*[10], Euphorbia subgenus species *E. tirucalli* and the Chamaesyce subgenus species *E. pulcherrima*[11], providing a valuable source for plant genome research and gene discovery. Many *Euphorbia* species produce bioactive diterpenoid compounds such as ingenol mebutate (*E. peplus*), resiniferatoxin (*E. resinifera*) or jatrophanes[12–14]. Several triterpenes have also been isolated from the *Euphorbia* genus, including linear and cyclic triterpenes such as peplusol, lanosterol, cycloartenol, 24-methylenecycloartenol and others[15,16].

Peplusol (**1**) (Fig. 1A) is an unusual linear triterpene alcohol produced by a few members of the *Euphorbia* genus including *E. peplus, E. lateriflora, E. guyoniana* and *E. sikkimensis*[15,17–19]. Peplusol levels reaching 5 mg/g have been reported in *E. peplus* latex[20], where it may play a role determining physical properties of this milky white fluid[15]. Previous experiments have also shown moderate antifungal activities of peplusol against agricultural phytopathogenic fungi[21]. Other squalene-related linear triterpenes have been described as natural products, such as botryococcenes and bisfarnesyl ether from green algae *Botryococcus braunii* (Fig. 1A)[22] and dehydrosqualene from *Staphylococcus aureus*[23]. Squalane, a fully hydrogenated derivative of

[1]Centre for Novel Agricultural Products, Department of Biology, University of York, Heslington, York YO10 5DD, UK. [2]Plant Sciences for the Bioeconomy, Rothamsted Research, Harpenden AL5 2JQ, UK. ✉e-mail: ian.graham@york.ac.uk

**A**

**B**

Fig. 1 | **Proposed formation of selected linear triterpenes. A** Examples of linear triterpenes found in nature. **B** Proposed scheme for peplusol and squalene formation from farnesyl diphosphate (FPP) precursor.

squalene **(2)** (Fig. 1A), is widely used in cosmetics and skin care. Linear triterpenes such as squalene and botryococcene have proven to have efficacious immunomodulating properties with the former now used as an essential component of nanoemulsion vaccine adjuvants, such as MF59[24]. Squalene, originally sourced from shark liver oil and plant oils (mostly olive oil) is now a target for industrial biotechnology to deliver a more sustainable supply. The presence of a C1 hydroxyl group on peplusol distinguishes its structure from these other linear squalene-like triterpenes, making it amenable to chemical- or enzymatic modifications that could introduce new or improve existing functionalities.

The mechanism of squalene biosynthesis by squalene synthase, a divalent metal-ion-dependent enzyme, has been well characterized[25] (Fig. 1B). The catalytic process involves a three step reaction: firstly, two FPP molecules are condensed to form presqualene diphosphate (PSPP), an intermediate with a cyclopropane ring structure; secondly, opening of the PSPP cyclopropyl group yields an intermediate carbonium ion which rearranges and is then quenched by NADPH to yield squalene. In the presence of $Mg^{2+}$ and NADPH, FPP is efficiently converted into squalene, whereas in the absence of NADPH reducing

power and the presence of $Mn^{2+}$ ions, human squalene synthase yields dehydrosqualene (Fig. 1A)[25]. In contrast, the biosynthesis of peplusol is not understood, but a six-step chemical synthesis, starting from methyl acetoacetate has been achieved[17].

A genome wide analysis of squalene synthase gene homologues in monocot and dicot plant species has shown that all plant squalene synthases are derived from a single ancestral gene and with copy number variation in the genomes that are derived from lineage-specific duplications[26]. Paralogues of the squalene synthase gene present in plant genomes often show diverse expression patterns, leading to the speculation that the duplicated copies of squalene synthase genes might have contributed to the evolution of different physiological roles or novel functions in specific lineages[26]. It has also been proposed that squalene synthase-like enzyme diversification in green algae gave rise to botryococcenes and bisfarnesyl ether synthesis via gene duplication and neofunctionalization some 500 million years ago[22].

Here, we describe the discovery and functional characterisation of two peplusol synthases from the *Euphorbia* genus and their evolutionary relationship with squalene synthase.

## Results and discussion

### Identification and functional characterisation of peplusol and squalene synthases from *E. peplus*

We hypothesised that the mechanism for peplusol formation would be similar to that of the well characterised squalene formation by squalene synthase, as depicted in Fig. 1B, with synthesis of both beginning with head-to-head condensation of two FPP molecules. The crucial difference between the two reaction end-products is the lack of the cyclopropane ring formation in peplusol synthesis that is present in presqualene diphosphate (PSPP), a squalene biosynthesis intermediate. We propose that as a result of this difference peplusol diphosphate (PPP) is not formed directly from PSPP. In this route, the NADPH quenching agent is not essential for peplusol formation as it derives from the hydrolysis of peplusol diphosphate.

On the basis of the proposal presented in Fig. 1B, we searched for squalene synthase-like enzymes in the *E. peplus* genome assembly[9] and identified two adjacent squalene synthase sequences annotated on chromosome 7 (Supplementary Fig. 1A). These genes, named *EpSS-L1 and EpSS-L2*, are in head to head orientation, and share 79% nucleotide and 72% amino acid identity, suggesting a tandem gene duplication. Analysis of the expression profile of *EpSS-L1* and *EpSS-L2* in an existing RNAseq dataset[20] revealed that they are both highly expressed in latex - a milky sap secreted by laticiferous tissues. However, *EpSS-L1* expression is latex-specific whereas expression of *EpSS-L2* is more ubiquitous (Supplementary Fig. 1B).

We amplified both *EpSS-L1* and *EpSS-L2* coding sequences from *E. peplus* stem cDNA, and cloned these into a pEAQ-HT vector under the control of the CaMV-35S promoter, for transient expression in *N. benthamiana*. *EpSS-L1* and *EpSS-L2* were transiently expressed in *N. benthamiana* leaves either with or without a truncated *3-hydroxy-3-methylglutaryl coenzyme A (HMG-CoA) reductase* sequence (*HMGRt*) from *A. thaliana*. *HMGR* encodes the key enzyme involved in IPP biosynthesis via the mevalonate pathway and it was previously shown that overexpression of truncated *HMGR* increases supply of the FPP precursor resulting in enhanced levels of squalene and other triterpenes in *S. cerevisiae* [27] and *N. benthamiana* [28]. Truncation of the N-terminal membrane-binding region was shown to remove feedback control of HMGR activity[29]. *N. benthamiana* transformed with previously characterised *A. thaliana Squalene Synthase* (*AtSS*)[30] and pEAQ-HT empty vector (*EV*) were used as experimental controls in transient expression assays in infiltrated leaves with both resulting in an unknown *N. benthamiana* compound detected in extracted ion chromatograms (Fig. 2A, peak 3). Compound (**3**) was extracted from infiltrated leaves of *N. benthamiana*, purified and characterized by NMR, revealing its identity as 2,3-oxidosqualene (Supplementary Fig. 2A, Supplementary Note 1). This compound can be formed by oxidation of squalene by squalene monooxygenase[31,32]. We assume 2,3-oxidosqualene is produced by an endogenous squalene monooxygenase in *N. benthamiana* to generate the substrate for various oxidosqualene cyclases involved in essential sterol biosynthesis. The same assay conditions show that *EpSS-L1* produces peplusol with or without the *A. thaliana HMGRt*, indicating it encodes a functional peplusol synthase (Fig. 3A, B). High resolution mass spectrometry (HRMS) data for both peplusol and 2,3-oxidiosqualene produced in *N. benthamiana* heterologous host by overexpression of EpSS-L1 and EpSS-L2 are nearly identical with the NMR-verified standards (Supplementary Fig. 3A and 3B). Expression of *EpSS-L1* along with *HMGRt* increases production of peplusol 35-fold compared to in the absence of *HMGRt*, reaching over 2.5% of leaf dry weight (Fig. 2B). There was no peplusol production detected in the *EV, EpSS-L2* or the *AtSS* treatments (Figs. 2A and 2B). Transient expression of *EpSS-L1* in combination with *HMGRt* significantly reduced the level of squalene (12-fold) in *N. benthamiana* leaves (Fig. 2C). Levels of 2,3-oxidosqualene were also significantly reduced when *EpSS-L1* was transiently expressed with (5-fold) and without (10-fold) *HMGRt* (Figs. 2D and 2E). These results may be due to EpSS-L1

competing with the endogenous *N. benthamiana* squalene synthase for FPP precursor but this would need to be confirmed by assaying FPP levels. There is no significant change in squalene levels when *AtSS* or *EpSS-L2* are expressed transiently with *HMGRt*, (Fig. 2C). However levels of 2,3-oxidosqualene significantly increased when *AtSS* and *EpSS-L2* are expressed without *HMGRt* (4-fold and 6.7-fold, respectively, Fig. 2D), which indicates *EpSS-L2* encodes a functional squalene synthase.

To ensure that *N. benthamiana* host enzyme activities were not confounding functional characterization of *EpSS-L1* and *EpSS-L2*, both candidate genes were also tested in a *Saccharomyces cerevisiae* heterologous host system alongside previously characterized *A. thaliana squalene synthase (AtSS)*[30] and empty vector (EV) controls. Codon-optimised plant sequences were cloned into the pBEVY-L vector, under the control of a strong *Gal10* promoter and transformed into *S. cerevisiae* wild type (CEN.PK2-1C) background (*tHMG1*-minus) as well as into a strain with chromosomal integration of a truncated *HMG-CoA reductase* (*tHMG1* plus). Expression of codon-optimised *EpSS-L1* in *S. cerevisiae* confirmed the finding from *N. benthamiana* transient expression that expression of this gene results in production of peplusol (Fig. 3A). HRMS data for peplusol produced in *S. cerevisiae* heterologous by overexpression of EpSS-L1 are nearly identical with the NMR-verified standards, as shown in Supplementary Fig. 3A. Co-expression of *EpSS-L1* with the truncated *S. cerevisiae HMG-CoA reductase* (*tHMG1* plus background) increased peplusol production 3-fold, reaching 23 mg/L in resulting cell pellets. These results demonstrate that most of the peplusol produced is retained in the *S. cerevisiae* cells as the concentration in cell pellets is 17-fold higher than in whole cultures or media (Fig. 3A). There was no peplusol production detected in *S. cerevisiae* expressing *EV, EpSS-L2* or *AtSS* (Fig. 3A). Expression of *EpSS-L1* also significantly elevates squalene levels in *S. cerevisiae* however, the effect is not as strong as in the case of overexpression of *AtSS* and *EpSS-L2* (Fig. 3B). Expression of *EpSS-L1* on squalene levels is also much more evident in the *S. cerevisiae* that lacks *tHMG1* (10-fold increase) than in the *tHMG1*-plus background (1.5-fold increase). Discrepancies between squalene production driven by *EpSS-L1* in *N. benthamiana* and *S. cerevisiae* could be explained by a much more dynamic pool of FPP/squalene present in plants than in yeast cells. It is noteworthy that overexpression of *AtSS* has resulted in the significant increase of squalene levels in *N. benthamiana*, even when combined with *HMGRt* (Fig. 2C)

Overexpression of *EpSS-L2* and *AtSS* in the absence of *tHMG1* result in significantly increased squalene levels (220- and 540-fold, respectively). In the presence of *tHMG1* the effect of both *EpSS-L2* and *AtSS* is much less marked as even the empty vector control accumulates significant levels of squalene presumably due to increased metabolic flux into FPP and this being available to the *S. cerevisiae* squalene synthase (Erg9p). (Fig. 3B). Results from heterologous expression in *N. benthamiana* and *S. cerevisiae* strongly indicate that *EpSS-L2* is a functional squalene synthase.

### Identification and functional characterisation of peplusol and squalene synthases from *E. lateriflora*

Peplusol production has been reported in a few other *Euphorbia* species, including *E. lateriflora*[17]. We grew verified specimens of *E. lateriflora* alongside *E. peplus* in controlled glasshouse conditions (Supplementary Fig. 4B). Metabolite profiling of latex revealed that *E. lateriflora* peplusol levels are almost double those of *E. peplus* (Supplementary Fig. 4A and 4D). It is also noteworthy that *E. lateriflora* latex contains other cyclic triterpenes, previously described in *E. peplus*[15], such as lanosterol (**4**), cycloartenol (**5**) and 24-methylenecycloartenol (**6**) (Supplementary Fig. 4A and 4D). We extracted and purified peplusol from *E. lateriflora* and *E. peplus* and validated their structures by NMR. We confirmed that both structures have identical $^1H$ NMR spectra and also HRMS data (Supplementary Fig. 4C and 2B, Supplementary Note 1).

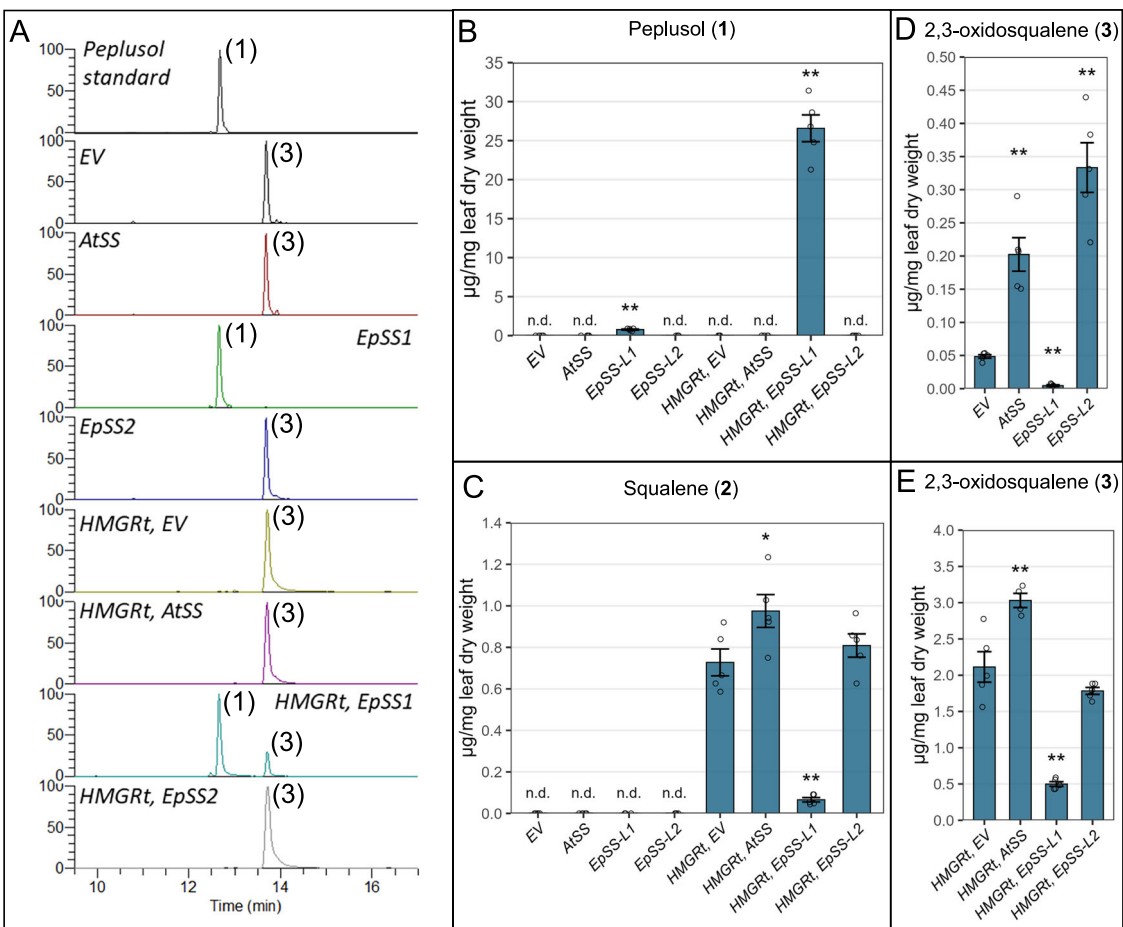

**Fig. 2 | Functional characterisation of *EpSS-L1* and *EpSS-L2* in *N. benthamiana*.** **A** Four week old *N. benthamiana* plants infiltrated with: Empty Vector (*EV*), *A. thaliana* Squalene Synthase (*AtSS*, GenBank accession BT003419.1), *E. peplus* Squalene Synthase-like1 (*EpSS-L1*, GenBank locus tag M5689_021805), *E. peplus* Squalene Synthase-like2 (*EpSS-L2*, GenBank locus tag M5689_021806) with and without truncated *A. thaliana* HMG-CoA reductase (*HMGRt*, GenBank accession J04537.1). Infiltrated leaves were extracted and analysed by LC-MS in parallel with *E. peplus* latex-purified peplusol standard. Extracted Ion Chromatograms (EIC) shown for *m/z* 427.4 showing peplusol (**1**) and 2,3-oxidosqualene (**3**) **B** Peplusol levels were quantified by LC-MS in *N. benthamiana* leaves infiltrated with gene combinations as on panel A, against a standard curve generated from purified peplusol.

n.d.: not detectable, Error bars: SEM (*n* = 5). **C** Squalene levels were quantified by GC-MS in *N. benthamiana* leaves infiltrated with gene combinations as on panel A, n.d.: not detectable, Error bars: SEM (*n* = 5, where n is the number of biological replicates). Statistically significant (one-sided *t*-test) changes between control (*EV*) and candidate genes indicated by asterisks (*: *p*-value < 0.05, **: *p*-value < 0.01). **D**, **E** 2,3-oxidosqualene levels were quantified by LC-MS in *N. benthamiana* leaves infiltrated with gene combinations as on (**A**) against a standard curve generated from purified 2,3-oxidosqualene. Error bars: SEM (*n* = 5, where n is the number of biological replicates). Statistically significant (one-sided *t*-test) changes between control (*EV*) and candidate genes indicated by asterisks (**: *p*-value < 0.01). Source data for (**B**–**E**) are provided as a Source Data file.

NMR and HRMS data for peplusol are also in agreement with previously published data[15].

As there was no genome or transcriptome data available for *E. lateriflora*, we performed RNAseq on total RNA extracted from *E. lateriflora* latex-containing stems, followed by de novo transcriptome assembly. Homology based searches using *EpSS-L1* and *EpSS-L2* coding sequences (CDS) allowed us to identify a full length CDS for a candidate peplusol synthase homologue, *ElSS-L1*, sharing 90% nucleotide and 89% amino acid identity with *EpSS-L1*. We also identified a candidate squalene synthase homologue, *ElSS-L2*, sharing 96% nucleotide and 95% aa identity with *EpSS-L2*. We cloned CDS for *ElSS-L1 and ElSS-L2* into a pEAQ-HT vector for transient expression in *N. benthamiana* as described above for *E. peplus*. *ElSS-L1* was able to drive peplusol production at levels similar to *EpSS-L1*, indicating it encodes a functional peplusol synthase (Fig. 4A). Co-expression of *AtHMGRt* increased production of peplusol by 20-fold, reaching 1.4% of leaf dry weight (Fig. 4A). Overexpression of *ElSS-L1* reduced level of squalene (26-fold) in *N. benthamiana* leaves when combined with *AtHMGRt*, similarly to *EpSS-L1* (Fig. 4B). Levels of 2,3-oxidosqualene were also strongly reduced, when *ElSS-L1* were overexpressed without

(20-fold) and with (7.5-fold) *AtHMGRt*, similarly to *EpSS-L1* (Fig. 4C). *ElSS-L2* overexpression did not yield any peplusol, however it significantly increased the level of 2,3-oxdiosqualene when expressed with and without *AtHMGRt* (Fig. 4C). Given the above result and high sequence homology with *EpSSL-2* which predominantly exhibits squalene synthase activity (Fig. 3) we suggest that *ElSS-L2* likely also encodes a squalene synthase. HRMS data for both peplusol and 2,3-oxdiosqualene produced in *N. benthamiana* heterologous host by overexpression of *ElSS-L1* and *ElSS-L2* are nearly identical with the NMR-verified standards, as shown in Supplementary Fig. 3A and 3B.

### Phylogenetic analysis of plant squalene synthases and peplusol synthases

The *EpSS-L1* and *ElSS-L1* genes that we have shown to exhibit peplusol synthase activity share a substantial degree of sequence similarity to squalene synthases (79% nucleotide and 72% amino acid identity). We used phylogenetic gene tree analyses to better understand the evolutionary origin of peplusol synthases in relation to squalene synthases in eudicot plants. This analysis shows that *EpSS-L1* and *ElSS-L1* likely evolved from the neofunctionalization of a duplicated copy of

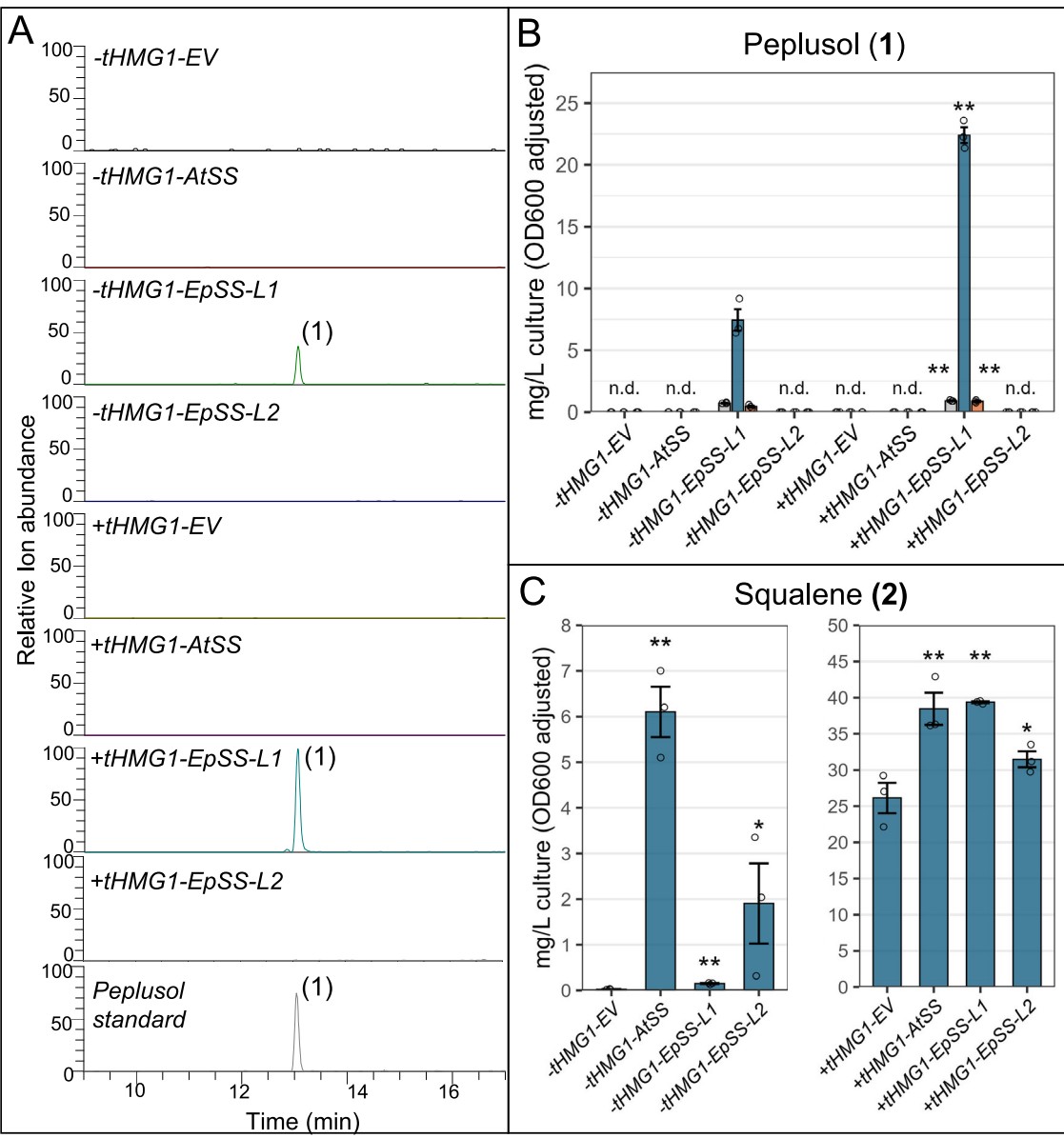

**Fig. 3 | Functional characterisation of *EpSS-L1* and *EpSS-L2* in *S. cerevisiae*. A** *S. cerevisiae* CEN.PK2-1C, wild type strain (-tHMG1) and *truncated HMG-CoA reductase 1*-expressing strain (+tHMG1) were transformed with either Empty Vector (EV, pBEVY-L) or pBEVY-L constructs for overexpression of: *A. thaliana Squalene Synthase (AtSS*, GenBank accession BT003419.1), *E. peplus Squalene Synthase-like1* (*EpSS-L1*, GenBank locus tag M5689_021805) and *E. peplus Squalene Synthase-like2* (*EpSS-L2*, GenBank locus tag M5689_021806). Empty Vector (*EV*), *A. thaliana Squalene Synthase (AtSS*, GenBank accession BT003419.1), *E. peplus Squalene Synthase-like1* (*EpSS-L1*, GenBank locus tag M5689_021805), *E. peplus Squalene Synthase-like2* (*EpSS-L2*, GenBank locus tag M5689_021806) with and without truncated *A. thaliana HMG-CoA reductase* (*HMGRt*, GenBank accession J04537.1). *S. cerevisiae* cultures were extracted and analysed by LC-MS in parallel with *E. peplus* latex-purified peplusol standard. Extracted Ion Chromatograms (EIC) shown for *m/z* 427.4 showing peplusol (**1**) in extracts derived from cell pellets (**B**) Quantification

of peplusol by LC-MS from the extracts shown on (**A**). Three independent transformants were grown in liquid cultures. Peplusol levels were quantified by LC-MS in: whole liquid cultures (grey bars), cell pellets (blue bars) and media (orange bars), against a standard curve generated from purified peplusol. n.d.: not detectable, Error bars: SEM ($n = 3$, where n is the number of biological replicates). Statistically significant (one-sided *t*-test) changes between *EpSS-L1* overexpressed in -tHMG1 and +tHMG1 backgrounds indicated by asterisks for whole liquid cultures, cell pellets and media (\*: *p*-value < 0.05, \*\*: *p*-value < 0.01). **C** Quantification of squalene by GC-MS. Three independent transformants were grown in liquid cultures. Squalene levels were quantified by GC-MS in cell pellets from the strains shown in panel A. Error bars: SEM ($n = 3$, where n is the number of biological replicates). Statistically significant (one-sided *t*-test) differences between control (EV) and candidate genes indicated by asterisks (\*: *p*-value < 0.05, \*\*: *p*-value < 0.01). Source data are provided as a Source Data file.

squalene synthase in a common ancestor in the *Euphorbia* subgenus Esula with the two peplusol synthases forming a sister relationship (Fig. 5). The long branch of this peplusol synthase clade may reflect an accelerated rate of nucleotide changes in the neofunctionalization process in the duplicated copy of squalene synthase. The gene tree position of the peplusol synthase clade and the presence of an orthologous copy of the peplusol synthase gene in the assembled genome of *E. tirucalli*[33] suggests that this gene duplication occurred in

the common ancestor of the *Euphorbia* genus before the separation of the Euphorbia subgenus species *E. tirucalli* and the Chamaesyce subgenus species *E. pulcherrima* from *E. peplus* and *E. lateriflora* 34.2 million years ago. This analysis is consistent with previous work showing that *E. peplus* and *E. lateriflora* shared a common ancestor 29.6 million years ago[34] (http://timetree.org). Our results indicate a monophyletic origin of the peplusol synthase gene after a *Euphorbia*-specific gene duplication event of the squalene synthase gene before

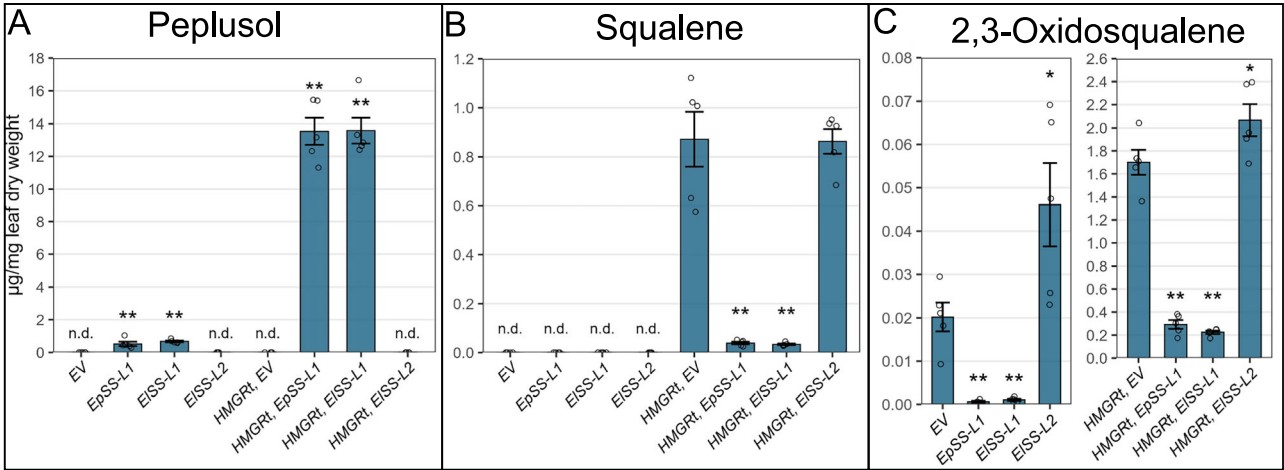

**Fig. 4 | Functional characterisation of *ElSS-L1* and *ElSS-L2* in *N. benthamiana*.** **A** Peplusol levels were quantified by LC-MS in *N. benthamiana* leaves infiltrated with: Empty Vector (*EV*), *E. peplus Squalene Synthase-like1* (*EpSS-L1*, GenBank locus tag M5689_021805), *E.lateriflora Squalene Synthase-like1* (*ElSS-L1*, GenBank accession PP978604), *E. lateriflora Squalene Synthase-like2* (*ElSS2*, GenBank accession PP978605), with and without truncated *A. thaliana HMG-CoA reductase* (*HMGRt*, GenBank accession J04537.1). n.d.: not detectable, Error bars: SEM ($n = 5$ where n is the number of biological replicates). **B** Squalene levels were quantified by GC-MS in *N. benthamiana* leaves infiltrated with gene combinations as on panel A. n.d.: not detectable, Error bars: SEM ($n = 5$ where n is the number of biological replicates). Statistically significant (one-sided *t*-test) changes between control (EV) and candidate genes indicated by asterisks (**: *p*-value < 0.01. **C** 2,3-oxidosqualene levels were quantified by LC-MS in *N. benthamiana* leaves with gene combinations as on (**A**). Error bars: SEM ($n = 4$ or 5, where n is the number of biological replicates). Statistically significant (one-sided *t*-test) differences between control (EV) and candidate genes indicated by asterisks (**: *p*-value, < 0.01, *: *p*-value, < 0.05). Source data are provided as a Source Data file.

the divergence of the subgenera (Fig. 5). The presence of the orthologous sequence in *E. tirucalli* implies that peplusol might have actually been occurring more widely in other species of the *Euphorbia* genus apart from *E. peplus*, *E. lateriflora*, *E. guyoniana* and *E. sikkimensis*[15,17–19]. We have not detected the peplusol synthase orthologues in either the reported genome of another Esula subgenus species *E. lathyris*[10] that diverged before the split of *E. peplus* and *E. lateriflora* 33.5 million years ago, or in the assembled genome of the Chamaesyce subgenus species *E. pulcherrima*[35]. This suggests both these species had inherited but lost the peplusol synthase gene and also lost the ability to synthesise peplusol in their separate lineages. We were able to source *E. lathyris* from nature and verified its identity using the conserved orthologue gene set (COS). Metabolite profiling of *E. lathyris* latex did not detect peplusol (Supplementary Fig. 4D), which is consistent with the lack of the peplusol synthase gene homologue in the genome of this species[10].

**Molecular evolution of peplusol synthase activity**

To investigate if it is possible to modify a squalene synthase enzyme such that it gains peplusol synthase activity we first looked for potentially important amino acid positions. Alignment of cDNA-predicted amino acid sequences for 32 functionally characterised squalene synthases from plant, algae, animal and fungi species with functionally characterised *E. peplus* and *E. lateriflora* peplusol synthases (*EpSS-L1 and ElSS-L1*, Supplementary Fig. 5A) revealed extremely high homology within four regions. These surround a centrally located cavity which constitutes the hydrophobic core of the human squalene synthase active site, crucial for its catalytic activity[25]. Alignment shows that two aspartate rich motifs (DXXED), involved in FPP substrate binding are fully conserved among peplusol and squalene synthases, consistent with the mechanisms presented in Fig. 1B. There are a small number of amino acid positions within the four regions, which are unique to the ElSS1 and EpSS1 protein sequence and conserved among all squalene synthases (Supplementary Fig. 4A). We identified six residues: M69, M208, T209, G294, Y298 and P313, located in three out of four conserved sequence regions as potentially important for differentiating peplusol from squalene synthase activity (Supplementary Fig. 5A).

We used cDNA-predicted amino acid sequences of peplusol (EpSS-L1, ElSS-L1) and putative squalene synthases (EpSS-L2 and ElSS-L2) to model their 3D protein structure using the AlphaFold 3 release of the Google DeepMind AI protein structure modelling tool, which is significantly improved in prediction of biomolecular interactions compared to previous versions[36]. The N- and C-terminal α-helices (the latter involved in anchoring squalene synthase enzymes in the endoplasmic reticulum) were reaching a per-atom confidence estimate (pIDDT) of 50-70 or 70-90, whereas the vast majority of each of the four structures was modelled with very high confidence (pIDDT > 90), which is also reflected by their high predicted template modelling (pTM) overall score (Supplementary Fig. 5B). Strong preservation of the squalene synthase fold evident for all four structures allowed us to overlay them with the PDB-deposited crystal structures of human squalene synthase complexed with a) presqualene pyrophosphate (PSPP) and $Mg^{2+}$ (Supplementary Fig. 6A–6C); and b) farnesyl *S*-thiolodiphosphate (FsPP, a non-cleavable FPP analogue) and $Mg^{2+}$ (Supplementary Fig. 5D–5E). Removing low(er) confidence modelled parts of ElSS-L1, EpSS-L1, ElSS-L2 and EpSS-L2 protein structures results in a very good overall structural alignment with both crystal structures of human squalene synthase (Supplementary Fig. 6 A and 6D). Close examination of six positions potentially important for the catalytic activity of peplusol synthase revealed that four of them (M69, M208, T209 and Y299) are located in close proximity (< 6 Å) of the FsPP and PSPP binding sites, with T209 being the closest to the cyclopropane ring present in PSPP (Supplementary Fig. 6B and 6C). It is also noteworthy that one of the positions, P313, is in a flexible JK-loop ([309]VKMRP[313]) proposed to be crucial for NADPH binding and reduction of the C1-C1' double bond in the dehydrosqualene intermediate[25](Supplementary Figs. 5A, 6B, and 6E). Substitution of R315 in EpSS-L2, which is a very conserved position in all functionally characterised squalene synthases (Supplementary Fig. 5A) by P313 in EpSS-L1 may suggest disruption of NADPH ability and the absence of a reducing step in the biosynthesis of peplusol, which is consistent with the model proposed in Fig. 1B.

We have investigated the effect on enzyme activity of swapping the six positions mentioned above (Figs. 6A and 6B) and between a functional *E. peplus* peplusol synthase (EpSS-L1) and squalene synthase

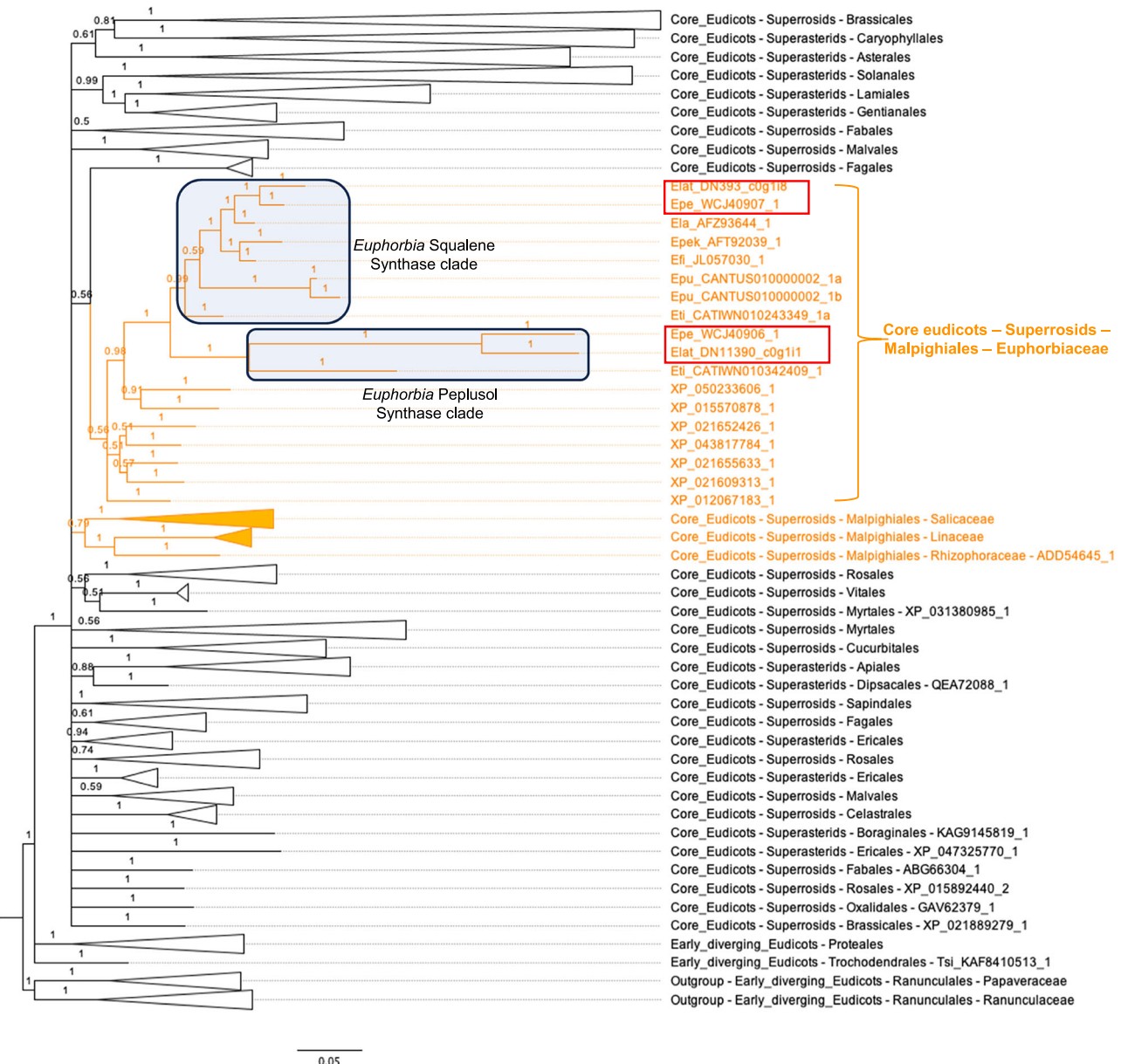

**Fig. 5 | Evolutionary origin of peplusol synthases inferred from the phylogenetic tree of homologues of squalene synthases of eudicots.** The phylogenetic tree of homologues of squalene synthases of eudicots were constructed using Bayesian inference. The posterior probabilities are shown on the branches and scale bar represents substitutions per nucleotide site. With the exception of the Malpighiales order, clades that contain species from the same order are collapsed and the names of the taxonomic order are labelled. Members in the Malpighiales order are highlighted in orange. All members from the Euphorbiaceae family are shown whereas other families are also collapsed and family names are indicated. A three or four letter prefix followed by an underscore is used as a species identifier for each gene from the *Euphorbia* species; including Efi_ (*E. fischeriana*), Ela_ (*E. lathyris*), Elat_ (*E. laterifolia*), Epe_ (*E. peplus*), Epek_ (*E. pekinensis*), Epu_ (*E. pulcherrima*), and Eti_ (*E. tirucalli*). The sister clades of *Euphorbia* peplusol synthases and squalene synthases are highlighted by shaded rectangles. Sequences heterologously expressed in *N. benthamiana* and/or *S. cerevisiae* highlighted in red box. See Supplementary Data 2 for the full list of the sequences used to construct phylogenetic trees with gen bank accessions.

(EpSS-L2) by creating two hexaswap constructs: *EpSS-L1to-L2_6AA* (M69F, M208T, T209N, G294T, Y298C, P313R) and *EpSS-L2to-L1_6AA* (F69M, T208M, N209T, T296G, C300Y, R315P). Plant sequences, codon optimised for yeast, were cloned into a pBEVY-L vector, under the control of a strong *Gal10* promoter and transformed into *S. cerevisiae* strain CEN.PK2-1Cas well as into CEN.PK2-1C plus a chromosomal integration of truncated *HMG-CoA reductase* (*tHMG1*). We found that strains expressing the *EpSS-L2to-L1* hexaswap construct were able to synthesise peplusol in both wild-type (*tHMG1* minus) and tHMG1 expressing (*tHMG1* plus) backgrounds at the levels reaching 40% of those produced by EpSS-L1 (Fig. 6C). Squalene production by the *EpSS-*

*L2to-L1* hexaswap dropped significantly (4-fold) when compared to the *EpSS-L2* in the absence of *tHMG1* (Fig. 6D) and are indistinguishable from the Empty Vector (EV) control, suggesting the six substitutions have abolished squalene synthase activity of EpSS-L2. *tHMG1 plus* strains expressing the *EpSS-L1to-L2* hexaswap produced significantly less peplusol than when *EpSS-L1* was expressed (4-fold reduction). The same mutations completely abolished peplusol production in a tHMG*1* minus background (Fig. 6C). The *EpSS-L1to-L2* hexaswap construct did not seem to gain any squalene synthase activity, as its squalene levels were comparable with *EV* control and *EpSS-L1* expressing strain in a *tHMG1* minus background (Fig. 6D).

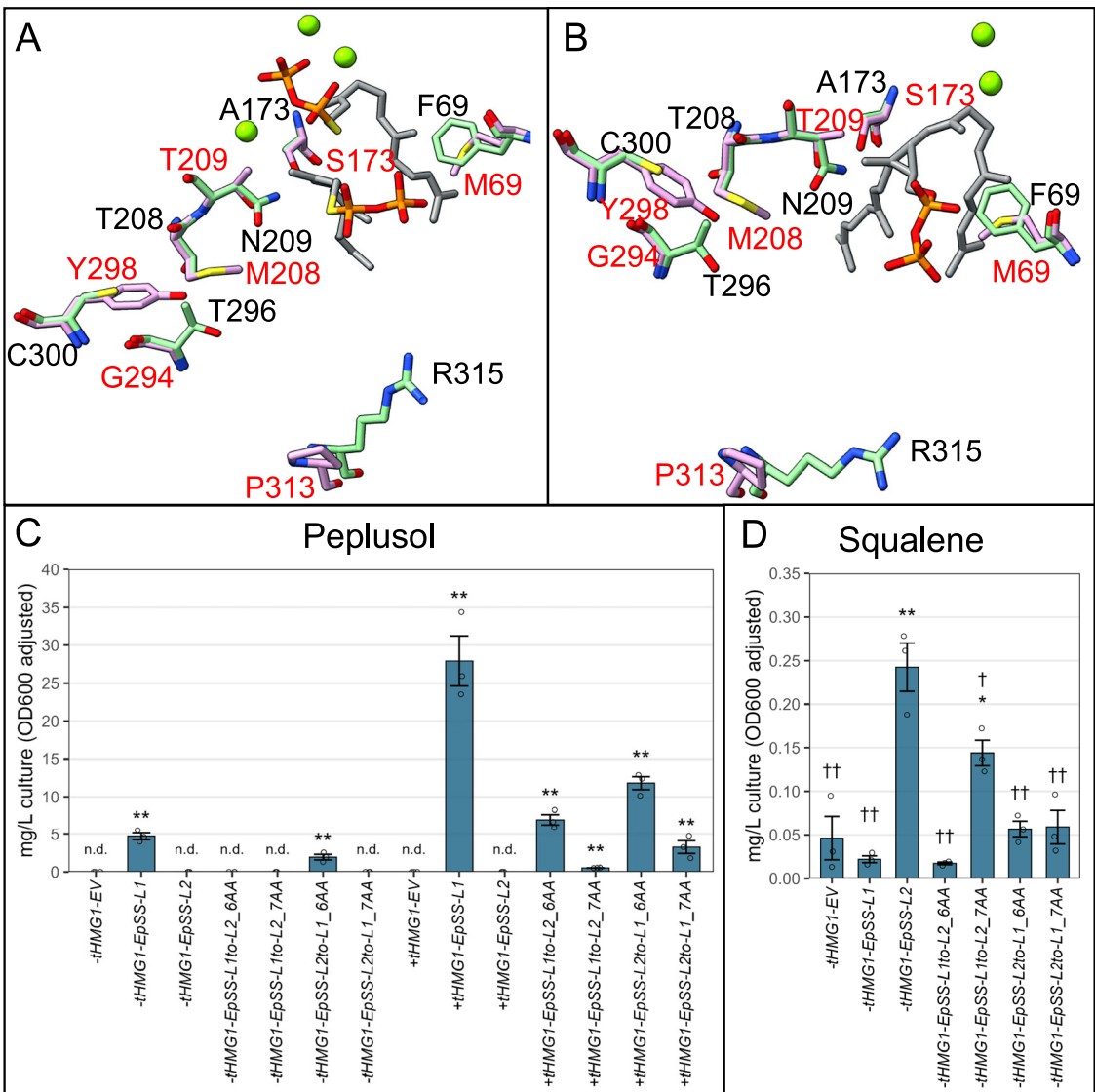

**Fig. 6 | Active site amino acid swaps convert squalene synthase into peplusol synthase and vice versa.** Impact of conserved active site hexa- and hepta- amino acid swaps on 3D protein structure of EpSS-L1 and EpSS-L2 were determined using Alpha Fold 3[36]. Highest ranked models (green for EpSS-L1, magenta for EpSS-L2) were overlaid with (**A**) PDB-deposited (3weh) structure human squalene synthase in complexes with presqualene pyrophosphate (PSPP, grey) and $Mg^{2+}$ (green spheres) or (**B**) PDB-deposited (3weg) structure human squalene synthase in complexes with farnesyl thiopyrophosphate (FsPP, a noncleavable FPP analogue, grey) and magnesium ion (grey) and $Mg^{2+}$ (green spheres). Seven positions identified as potentially important for peplusol synthase activity are shown as stick and wires and labelled for both EpSS-L1 (red) and EpSS-L2 (black). **C** *S. cerevisiae* CEN.PK2-1C, wild type strain (-tHMG1) and *truncated HMG-CoA reductase 1* -expressing strain (+tHMG1) were transformed with either: Empty Vector (EV, pBEVY-L) or pBEVY-L constructs containing *E. peplus Squalene Synthase-like2* (*EpSS-L2*), *E. peplus Squalene Synthase-like2 hexaswap* (*EpSS-L2to-L1_6AA*: F69M, T208M, N209T, T296G, C300Y and K318P), *E. peplus Squalene Synthase-like2*

*heptaswap* (*EpSS-L2to-L1_7AA*: F69M, T208M, N209T, T296G, C300Y, K318P and A173S), *E. peplus Squalene Synthase-like1* (*EpSS-L1*), *E. peplus Squalene Synthase-like1 hexaswap* (*EpSS-L1to-L2_6AA*: M69F, M208T, T209N, G294T, Y298C, P313R) and *E. peplus Squalene Synthase-like1 heptaswap* (*EpSS-L1to-L2_7AA*: M69F, M208T, T209N, G294T, Y298C, P313R and S173A). Three independent transformants were grown in liquid cultures. Peplusol in cell pellets was quantified by LC-MS. n.d.: not detectable, Error bars: SEM ($n = 3$ where n is the number of biological replicates). **D** Squalene levels in cell pellets was quantified by GC-MS from a subset of the -tHMG1 strains shown in panel C. Error bars: SEM ($n = 3$, where n is the number of biological replicates). Statistically significant (one-sided $t$-test) differences between control (EV) and candidate genes indicated by asterisks on (**C, D**) (*: $p$-value < 0.05, **: $p$-value < 0.01). Statistically significant (one-sided $t$-test) differences between strain overexpressing *E. peplus Squalene Synthase-like2* (*-tHMG1-EpSS-L2*) and other strains indicated by daggers on (**D**) (†: $p$-value < 0.05, ††: $p$-value < 0.01). Source data are provided as a Source Data file.

In a search for further conserved amino acid positions potentially allowing introduction of squalene synthase activity to *EpSS-L1* we identified another position, (S173 in EpSS-L1 and ElSS-L1 and A173 in EpSS-L2 and ElSS-L2) located in close proximity (< 6 Å) of the FsPP and PSPP binding sites, also close to the cyclopropane ring present in PSPP (Fig. 6A and 6B, Supplementary Fig. 6C and 6E). Protein sequence alignment revealed the A173 position is also very conserved among functionally characterised squalene synthases (Supplementary

Fig. 5A). Following the strategy described above, we created two heptaswaps constructs adding S173A to the six amino acid swaps described above, in the *S. cerevisiae tHMG1* minus and *tHMG1* plus backgrounds. The *EpSS-L1toL2_7AA* (M69F, M208T, T209N, G294T, Y298C, P313R and S173A) heptaswap construct gained significant squalene synthase activity when overexpressed in the CEN.PK2-1C, *tHMG1* minus background (Fig. 6D), whereas the same seven substitutions drastically reduced the peplusol synthase activity (50-fold

reduction) in CEN.PK2-1C *tHMG1* plus background (Fig. 6C). The *EpSS-L2to-L1* heptaswap construct (*EpSS-L2to-L1_7AA*: F69M, T208M, N209T, T296G, C300Y, K318P and A173S) does still synthesise peplusol but only in the tHMG1 expressing background and at levels significantly lower than those obtained for the *EpSS-L2to-L1* hexaswap (Fig. 6C). Finally, squalene production by the *EpSS-L2to-L1* heptaswap dropped significantly (4-fold) when compared to the *EpSS-L2* in a tHMG1 minus background and is not distinguishable from the empty vector (EV) control, suggesting the seven substitutions have abolished squalene synthase activity of EpSS-L2 (Fig. 6D). Results of expression of EpSS-*L2to-L1* and *EpSS-L1to-L2* hexa- and heptaswaps in *tHMG1*-plus background were not conclusive (Supplementary Fig. 7) as the effects are presumably masked again by increased activity of the endogenous squalene synthase (*Erg9p*) on elevated pools of FPP.

These results show that it is possible to convert squalene synthase to peplusol synthase and vice-versa using active site amino acid swapping. These findings are consistent with the phylogenetic analysis, which suggests that peplusol synthase activity most likely evolved from squalene synthase by gene duplication and neofunctionalization. It also opens the possibility of converting squalene synthase activity towards the synthesis of new-to-nature linear triterpenes with potential use as skin-emollients or vaccine-adjuvants.

### In vitro protein activity assays for AtSS, EpSS-L1, EpSS-L2, EpSS-L1to-L2 and EpSS-L2to-L1 heptaswap mutants

To further confirm the in vivo determined enzyme activities of the native and modified triterpene synthases, in vitro activity assays were performed using crude protein lysates from *E. coli* strains expressing the respective gene constructs. Previous work used this system to assay *AtSS*[30]. To overcome the challenge of low levels of soluble expressed protein in *E. coli* we truncated the AlphaFold3 predicted C-terminal α-helices as well as the first 10-amino acids from the N-terminus which were predicted to represent un-structured fragments of modelled protein structures (Supplementary Fig. 5B). Results of in vitro assays using crude lysates from *E. coli* expressing the native and modified triterpene synthases under a range of reaction conditions are summarized in Table 1, and Supplementary Figs. 8 and 9.

Squalene (**2**) was the main product of the in vitro assays on both AtSS and EpSS-L2 proteins crude lysates when tested at pH 6.5 and 7.5, with peak retention time and *m/z* profile closely matching those for the squalene standard (Table 1, Supplementary Fig. 8A-D and 9). Considerably higher squalene levels were detected in both AtSS and EpSS-L2 assays when NADPH or NADP+ and Mg2+ were present in the reaction, compared to assays using Mg2+ alone (Table 1, Supplementary Fig. 8A–D and 9). These results are consistent with the previously shown co-factor requirement for the formation of squalene catalyzed by various squalene synthases[25,30,37,38] and confirm our findings using *N. benthamiana* and *S. cerevisiae* that EpSS-L2 encodes a functional squalene synthase.

Peplusol (**1**) was the main product of in vitro assays on EpSS-L1 with the peak retention time and *m/z* profile closely matching those for the peplusol standard (Table 1, and Supplementary Fig. 8E, 8F, and 9). Levels of peplusol were highest when NADPH plus Mg2+ were used as co-factors and are similar at pH 6.5 and 7.5 (Table 1, and Supplementary Fig. 8E and 8F). However, maximum levels of peplusol produced in the in vitro assays are ~100-fold lower than squalene levels in the AtSS and EpSS-L2 assays (Table 1, and Supplementary Fig. 8E and 8F) which is in contrast to similar production levels of squalene and peplusol by EpSS-L2 and EpSS-L1 in the *S. cerevisiae* platform (Fig. 3). This suggests that the in vitro assay conditions we have used for peplusol synthase require further optimisation. Formation of small amounts of squalene (**2**) by EpSS-L1 was observed in some assay conditions, for example at pH 6.5 with NADP+ plus Mg2+ or with Mg2+ alone (Table 1, and Supplementary Fig. 8E). These in vitro results are in agreement with our findings from heterologus expression of EpSS-L1 in *N. benthamiana* and *S. cerevisiae* that EpSS-L1 encodes peplusol synthase activity with minor squalene synthase activity. Interestingly, EpSS-L1 in vitro assays also produced an additional product (**7**) with a very close retention time to squalene, however the *m/z* profile differs significantly from the squalene standard (Table 1, and Supplementary Fig. 8E, 8F, and 9).

In vitro assays on the EpSS-L1to-L2 heptaswap mutant protein produced both peplsuol (**1**) and squalene (**2**) across several experimental conditions (Table 1, and Supplementary Fig. 8G and 8H), with squalene being the major product in the presence of NADPH plus Mg2 +. These results are entirely consistent with those from heterologous expression of EpSS-L1to-L2 heptaswap mutant in *S. cerevisiae*. We have been unable to detect any peplusol from the in vitro assays on the EpSS-L2to-L1 heptaswap mutant protein and squalene levels are 400-fold reduced compared to EpSS-L2 (Table 1, and Supplementary Fig. 8I and 8J). These results are inconsistent with the heterologous

## Table 1 | Products of in vitro activity assays on peplusol and squalene synthases and heptaswap mutants

| | AtSS | | | EpSS-L2 | | | EpSS-L1 | | | EpSS-L1toL2 | | | EpSS-L2to-L1 | | | Empty vector | | |
|---|---|---|---|---|---|---|---|---|---|---|---|---|---|---|---|---|---|---|
| | 1 | 2 | 7 | 1 | 2 | 7 | 1 | 2 | 7 | 1 | 2 | 7 | 1 | 2 | 7 | 1 | 2 | 7 |
| **pH 6.5** | | | | | | | | | | | | | | | | | | |
| NADPH, Mg²⁺, FPP | - | ++++ | - | - | ++++ | - | ++ | - | ++ | ++ | ++ | - | - | + | - | - | - | - |
| NADP +, Mg²⁺, FPP | - | ++++ | - | - | +++ | - | + | ++ | + | + | - | - | - | - | - | - | - | - |
| Mg²⁺, FPP | - | +++ | - | - | +++ | - | + | ++ | + | + | - | - | - | - | - | - | - | - |
| FPP | - | - | - | - | - | - | ++ | - | - | - | - | - | - | + | - | - | - | - |
| NADPH, Mg²⁺ | - | - | - | - | - | - | - | - | - | - | - | - | - | - | - | - | - | - |
| **pH 7.5** | | | | | | | | | | | | | | | | | | |
| NADPH, Mg²⁺, FPP | - | ++++ | - | - | ++++ | - | ++ | + | ++ | + | ++ | - | - | - | - | - | - | - |
| NADP +, Mg²⁺, FPP | - | ++++ | - | - | ++++ | - | ++ | + | + | ++ | ++ | - | - | + | - | - | - | - |
| Mg²⁺, FPP | - | +++ | - | - | ++ | - | + | - | - | + | - | - | - | - | - | - | - | - |
| FPP | - | +++ | - | - | - | - | ++ | - | - | - | - | - | - | - | - | - | - | - |
| NADPH, Mg²⁺ | - | - | - | - | - | - | - | - | - | - | - | - | - | - | - | - | - | - |

**1**: peplusol (retention time – 22.86 min); **2**: squalene (retention time – 22.04 min); **7**: unknown (retention time – 21.92 min). Signal intensity for *m/z* 69 ions: "+++"-very high (1.1E06-4E06); "+++"-high (9E04-7.7E05); "++"-low (1E04-7.8E04); "+"-very low (3.2E03-9.8E03); "-"-not detectable (0) as detailed in Supplementary Fig. 8.

expression of EpSS-L2to-L1 heptaswap mutant in *S. cerevisiae* which introduced peplusol synthase whilst retaining squalene synthase activity (Fig. 6). We can only speculate that the introduced amino acid changes in the EpSS-L2to-L1 heptaswap mutant require as yet untested in vitro assay conditions.

Finally, we didn't observe peplusol (**1**), squalene (**2**), or unknown compound (**7**) from in vitro assays across all conditions on *E. coli* transformed with empty vector controls, even when very low intensity signals from *m/z* 69 ions were employed (Table 1, and Supplementary Fig. 8K and 8 L). As expected FPP substrate is essential for squalene production in all assays. In a few cases FPP alone was sufficient for production of either squalene or peplusol and we assume this is due to low levels of *E.coli* derived co-factors in the crude lysates.

In conclusion, squalene, a common precursor of all plant, fungal and animal sterols, is a high value natural product used by skin-care and pharmaceutical industries. Peplusol, a linear triterpene similar to squalene, occurring naturally in a very few members of the *Euphorbia* genus of flowering plants, has potential uses in crop-care (antifungal) as well as skin-care and pharmaceutical industries. Genomic and transcriptomic approaches described herein have led to the identification of two functional peplusol synthases: *EpSS-L1* and *ElSS-L1* from *E. peplus* and *E. lateriflora*, respectively, which share a high degree of sequence homology with functionally characterised plant squalene synthases. We present evidence for the evolution of peplusol synthase from squalene synthase via gene duplication and neofunctionalization, which has most likely occurred in a common ancestor of the *Euphorbia* genus before the separation of the Euphorbia and the Chamaesyce subgenus species, around 34.2 million years ago. We have used active site amino acid swaps to convert a squalene synthase into a peplusol synthase (and vice versa). This has pinpointed the key residues in determining product profile and highlights the feasibility of neofunctionalization. Based on this functional characterisation, we hereinafter refer to *EpSS-L1* as *EpPS*, *ElSS-L1* as *ElPS* and *EpSS-L2* as *EpSS*. *EpPS* is able to drive high level peplusol production in a *N. benthamiana* heterologous host (reaching over 2.5% leaf dry weight) and significant peplusol yields when overexpressed in *S. cerevisiae* (30 mg/L culture), opening up a synthetic biology route for peplusol production. Our findings expand the knowledge, of very rare non-squalene-dependent triterpene biosynthesis, beyond the recently discovered pathways operating in fungi[39].

## Methods

### Plant material

*E. peplus* seeds were obtained from All Rare Herbs (Australia). *E. lateriflora* cuttings were obtained from Botanic Garden Meise, Belgium (specimen GH-0-BR-2014067883, collected south of Oduponkpeke, Accra Plains, Ghana). Four *E. lathyris* plants were collected from the wild (Heslington, York, UK, location N53.94292547361107, E-1.0491734918701132), and grown in a glasshouse for six weeks (February-March 2024) using natural daylight and temperature. *E. peplus* plants were grown under 16 h/8 h light/dark and 21 ( + /−3) °C/17 ( + /−3) °C day/night regime for 8 weeks. Latex samples were collected at the flowering stage, by cutting main stems with a scalpel and collecting white sap by pipette into an Eppendorf tube (5-50 μL per plant) followed by freezing in liquid $N_2$ ($LN_2$). *E. lateriflora* plants were propagated via cuttings and grown under 12 h/12 h light/dark and 27 ( + /−3) °C/25 ( + /−3) °C day/night regime for 16 weeks. Plant latex was collected as above and latex-containing main stems were collected and frozen immediately in $LN_2$ for RNA extractions. *E. lathyris* latex was collected at the flowering stage as above.

### Preparation and identification of peplusol

A total of 4.8 mL of latex was harvested from 8-week old *E. peplus* plants, frozen in $LN_2$ and extracted with 480 mL of 100% ethyl acetate (Rathburn Chemicals, UK) over 5 days. Ethyl acetate was removed by rotary evaporation to yield 0.93 g of a dark yellow oily residue which was taken up in 20 mL of an *n*-hexane:ethyl acetate mixture (80:20). The extract was then applied to a 40 g Grace Resolve silica column and fractions collected using a 0–100 % ethyl acetate in hexane gradient, followed by isocratic 100% ethyl acetate and 100% methanol. This method yielded 10 mg of peplusol (**1**).

A total of 3 mL of latex was harvested from 16-week old *E. lateriflora* plants, frozen in $LN_2$ and extracted with 10 volumes of 100% ethyl acetate (Rathburn Chemicals, UK) over 5 days. Ethyl acetate was removed by rotary evaporation to yield 0.15 g of a yellow oily residue. The method described above was used to purify peplusol from dry extract, which yielded 9.5 mg of peplusol (**1**).

### Preparation and identification of 2,3-oxidosqualene

For transient gene expression, *N. benthamiana* leaves were transformed with *Agrobacterium tumefaciens* LBA4404 strain carrying a truncated version of *A. thaliana 3-hydroxy-3-methylglutaryl coenzyme A reductase 1* (*AtHMGRt*, GenBank accession AT1G76490, with 501 bp removed from 5′ end) and *AvGFP*, validated for use as a visual marker for tgene expression using the pEAQ-HT vector system[20,40]. Leaf material was harvested from 60 transformed *N. benthamiana* plants, freeze dried and extracted with 15 volumes of 100% ethyl acetate over 5 days. The extract (2.1 g) was then applied to a 40 g Grace Resolve silica column and fractions collected using a 0-100 % ethyl acetate in hexane gradient, followed by isocratic 100% ethyl acetate and 100% methanol. Fractions containing 2,3-oxidosqualene (**3**) were further purified using C18-3.5 μM 250 × 10 mm preparative reversed-phase HPLC column using a 20-60% gradient of 100% methanol with 0.2% formic acid in 80% methanol:water with 0.2% formic acid, followed by isocratic 60% acetone in 40% methanol with 0.2% formic acid. Analysis using this method yielded 30 mg of 2,3-oxidosqualene (**3**).

### DNA extraction from *E. lathyris* and species validation using conserved orthologue set

Genomic DNA was extracted from 100-150 mg of flash-frozen *E. lathyris* young leaves using a modified CTAB protocol. PCR primers (Supplementary Table 1) were designed to amplify fragments of a conserved orthologue set (COS)[41,42] gene set *Agt1*, *AroB*, *At1O3* and *Gbssi*. *E. lathyris* genomic regions covering ~500 bp upstream and downstream of the COS genes were extracted from the NCBI *E. lathyris* genome assembly under accession GCA_963576675.1 (https://www.ncbi.nlm.nih.gov/datasets/genome/GCF_963576675.1/). PCR was performed using PCR Bio Taq polymerase following manufacturers' protocol and all amplicons from each of the individuals were verified by Sanger sequencing.

### Metabolite profiling of *Euphorbia* species latex

Latex samples collected from stems of four individuals of *E. peplus*, *E. lathyris* were mixed with methanol, spun down at 20,000 g for 2 min at room temperature and 150 μL of the resultant supernatant analysed and quantified by LC-MS against phorbol 12-myristate 13-acetate (PMA) in a following way. A 2 μL aliquot was injected into an Acquity UPLC system (Waters, Elstree, UK) fitted with a Waters Acquity BEH C18, 2.1 mm x 100 mm, particle size 1.7 μm column (Waters Acquity cat. No. 186002352). Mass spectrometery was performed using a Thermo LTQ Orbitrap Classic (Thermo Fisher, UK), Obitrap FT-MS mass analyser with the following parameters: APIC ionization, FTMS 7500 resolution, scan range 100-1000 *m/z*, CID energy 40 (arb), MS1 full scan and data dependant MS2 from the most intense ion. Metabolites were eluted at 0.5 mL/min and 60 °C using following gradient program: 60% A and 40% B (isocratic), followed by a linear gradient to 100% B. (Solvent A: 5% methanol + 0.1% formic acid in water, Solvent B: 100% methanol + 0.1% formic acid) with the mass spectrometer settings as listed above. Data were acquired using Thermo Xcalibur software v. 2.1.0 SP1.1160 (Thermo Fisher). Four major triterpenoids were identified using either

commercial standards (lanosterol and cycloartenol) or purified and validated by NMR (peplusol and 24-methylenecyclartenol). Raw LC-MS data were analysed using Thermo Xcalibur software v. 4.0.27.10 (Thermo Fisher).

### RNA extractions from *E. lateriflora* and RNAseq analysis

cDNA library was prepared) from 1 µg total RNA, extracted from main stems of *E. lateriflora* with the RNAeasy kit (Qiagen, Hilden, Germany), using the NEBNext Ultra II Directional Library prep kit from Illumina in conjunction with the NEBNext® Poly(A) mRNA Magnetic Isolation Module (New England Biolabs), according to the manufacturer's instructions. A 13-minute fragmentation time was used when eluting mRNA from polyA magnetic beads. Amplification of the final library involved 9 cycles of PCR using the NEBNext multiplex Oligos (New England Biolabs). The sample was used for paired-end 150 base sequencing on an Illumina NovaSeq 6000 instrument and the resulting reads assembled into 201,887 contigs using the Trinity RNAseq de novo assembly software package[43]. TBLASTN searches were used to identify the *E. lateriflora* contigs that contain full length coding regions corresponding to both *EpSS-L1* and *EpSS-L2* genes.

### Candidate gene cloning and transient gene expression in *N. benthamiana*

cDNA was synthesised using total RNA from 100 ng of *E. peplus* latex or stems using Superscript II reverse transcriptase (Invitrogen, Carlsbad, CA) and random hexamer primers (Invitrogen, Carlsbad, CA). The open reading frame for each candidate gene was amplified using the primers detailed in Supplementary Table 2 and inserted into the pEAQ-HT expression vector using In-Fusion cloning tools (TaKaRa bio Inc. Kusatsu, Japan). In each instance, a 5'-AAAA-3' Kozak sequence was included immediately upstream of the start codon. In the case of both the *A. thaliana Squalene Synthase* (*AtSS*, GenBank accession AT4G34640) and the truncated version of *A. thaliana 3-hydroxy-3-methylglutaryl coenzyme A reductase 1 (AtHMGRt*, GenBank accession AT1G76490, with 501 bp removed from 5' end), the Genbank sequences were used to generate synthetic fragments containing an open reading frame with a 5' tail including a Kozak sequence (CTGTATATTCTGCCCAAATTCGCGAAAA) and a 3' tail (CCTTTAACTCTGGTTTCATTAAATT). The synthetic fragments were produced by Integrated DNA Technologies (Leuven, Belgium) and inserted into the pEAQ-HT expression vector[40] along with the *AvGFP* visual marker gene and transiently expressed in *N. benthamiana* leaves with the aid of *A. tumefaciens* as detailed above. Presence of *AvGFP* was used to guide tissue selection five days after *A. tumefaciens* infiltration, with all harvested material flash frozen in liquid nitrogen and subsequently analysed to produce data shown in Figs. 2 and 4.

### GC- and LC-MS analysis of plant material

Analytical methods for squalene, 2,3-oxidosqualene and peplusol extraction, detection and quantification were optimised from a previously reported method[44], which involved grinding plant material in a Retsch II homogenizer followed by extraction with 1 mL of ethyl acetate overnight. After centrifugation, ethyl acetate was removed by evaporation in a GeneVac personal evaporator (Genevac, Ipswitch, UK). The residue was re-dissolved in methanol and a 2 µL aliquot was injected into an Acquity UPLC system (Waters, Elstree, UK) fitted with an Accucore C30, 2.1 mm x 100 mm, particle size 2.6 µm column (Thermo Fisher cat. No. 27826-102130). Mass spectrometry was performed using a Thermo LTQ Orbitrap Classic (Thermo Fisher, UK), Orbitrap FT-MS mass analyser with the following parameters: APIC ionization, FTMS 7500 resolution, scan range 50-1200 $m/z$, CID energy 35 (arb), MS1 full scan and data dependant MS2 from the most intense ion. Metabolites were eluted at 0.35 mL/min and 40 °C using a linear gradient from: 99:1 solvent A: solvent B to 1:99 solvent A: solvent B over

21 min, followed by isocratic 1:99 solvent A: solvent B for 3 min and isocratic 99:1 solvent A: solvent B for 4 min (solvent A: 10 mM ammonium formate in methanol/water 60:40 + 0.1 % formic acid, solvent B: 10 mM ammonium formate in methanol/isopropanol 10:90 + 0.1 % formic acid) with the same mass spectrometer settings as listed above. Data were acquired using Thermo Xcalibur software v. 2.1.0 SP1.1160 (Thermo Fisher). A dilution series of *E. peplus*-purifed peplusol and *N. benthamiana*-purified 2,3-oxidosqualene was quantified in parallel using LC-MS to create a 7-point standard curve (0, 0.15625, 0.3125, 0.625, 1.25, 2.5 and 5 mg/mL for peplusol; 0, 0.03125, 0.0625, 0.125, 0.25, 0.5 and 1 mg/mL for 2,3-oxdidosqualene). The standard curves (with linear regression $R^2 \geq 0.998$) were used to calculate the amount of peplusol and 2,3-oxidosqualene in the plant extracts as presented in Figs. 2 and 4. Raw LC-MS data were analysed using Thermo Xcalibur software v. 4.0.27.10 (Thermo Fisher).

For squalene extraction plant material was ground in a Retsch II homogenizer and extracted in 500 µL of saponification solution (ethanol:water:KOH = 9:1:1, v:v:w) for 2 h at 65 °C with intermittent agitation. 250 µL of water was added to the samples before adding 500 µL of hexane. Samples were vortexed, centrifuged and 350 µL of the upper hexane phase was transferred to a glass vial and evaporated using a GeneVac personal evaporator (Genevac, Ipswich, UK). The dried extract was derivatised using a mixture of pyridine (60 µL), *N*-methyl-*N*-(trimethylSilyl)trifluoroacetamide (MSTFA, 30 µL) and trimethylsilyl chloride (TMS, 1 µL) for 1 h at 50 °C. A total of 1 µL of the derivatised extract was analysed by GC-MS using an Agilent 6890 Gas Chromatograph GC, (Agilent Technologies UK Ltd, Cheadle, UK) linked to a LECO Pegasus IV Time of Flight Mass Spectrometer TOF-MS, (LECO Instruments, Stockport, UK). The GC oven was fitted with a Restek Zebron ZB-5HT Inferno TM 30 M, capillary column (30 m, 0.25-mmID, 0.1 µm film thickness). Helium carrier gas was used at 1 mL/min constant flow and the transfer line temperature was set to 250 °C. The oven temperature was set at 80 °C for 5 min and then increased to 270 °C at a rate of 12 °C min⁻¹ followed by increase to 310 °C at a rate of 6 °C min⁻¹. Mass spectral data were acquired over the $m/z$ range of 50 to 500 in positive electron ionization mode at −70 eV with an acquisition rate of 20 spectra/second and a 5 min acquisition delay. Data were acquired using ChromaTOF software v. 4.50.8.0 (LECO Instruments, Stockport, UK) Squalene was quantified against an external standard curve, constructed from a commercial squalene standard (Sigma, cat. No. S3626), treated in the same way as extracts, using extracted ion $m/z$ 69, and presented in Figs. 2 and 4. Squalene peak from the plant extracts had nearly identical $m/z$ profile to the commercial squalene standard as shown on Supplementary Fig. 3C. ChromaTOF software output ".cdf" files were converted to ".raw" files and data analysed using Thermo Xcalibur software v. 4.0.27.10 (Thermo Fisher).

### Candidate gene cloning, expression in *S. cerevisiae*

Codon-optimised *EpSS-L1*, *EpSS-L2* and *AtSS*, (sequences listed in Supplementary Data 1) were synthesised as gBlock DNA fragments from IDT (Integrated DNA Technologies Inc.) with overhangs allowing direct insertion into a PmeI digested modified pBEVY-L vector, without PCR amplification, using In-Fusion cloning tools (TaKaRa Bio Inc., Kusatsu, Japan). Each codon optimised gene was assembled into a *Pme*I digested pBEVY-L vector (Addgene# 51225) under the control of the Gal10 promoter, the resulting constructs were propagated in Top10 *E. coli* cells (Invitrogen, USA) and confirmed by Sanger sequencing (Eurofins, Europe). Confirmed constructs were transformed into either the CEN.PK2-1C yeast wild type strain background (MATα ura3-52; trp1-289; leu2-3,112; his3Δ 1; MAL2-8c; SUC2, purchased from EUROSCARF, accession no-30000B) or the truncated *HMG1* overexpressing background using the standard lithium acetate transformation protocol[45]. The truncated *HMG1* was created by in vivo assembly of the *glyceraldehyde-3-phosphate dehydrogenase* promoter (*PTDH3*), *tHMG1*[46] and the cytochrome C oxidase

terminator (*TCYC1*) as DNA parts for genomic integration using CRISPR/cas9. The target guide sequence for genomic integration near the *ARS911* locus was identified using CRISPR too, version 4.99[47]. The top target sequence with zero off target count was selected for cloning. The guide sequence was cloned into *pCAS* vector (Addgene ID#60847)[48] and the construct confirmed by Sanger sequencing. The DNA parts were PCR amplified from CEN.PK2-1C genomic DNA with relevant 50 bp overlaps to the ends of each part (see primer sequences in Supplementary Table 3), to allow in vivo assembly and to enable integration by homologous recombination[49]. The *pCAS* plasmid with guide sequence and linear DNA parts were co-transformed into *S. cerevisiae* CEN.PK2-1C competent cells[48]. The transformation mix consisted of 1.0 µg of pCAS plasmid, 5.0 µg of linear repair DNA parts, 10 µL of ssDNA (Sigma-Aldrich D7656), 90 µL of yeast competent cells, and 900 µL of PLATE solution (50% poly-ethyleneglycol (PEG) 2000, 0.1 M lithium acetate, TE buffer (10 mM Tris-Cl, 1 mM EDTA pH 8.0) mixed in a 1.5-mL microcentrifuge tube. The mix was then incubated for 30 minutes at 30 °C followed by heat shock at 42 °C in a water bath. The supernatant was removed and cells were resuspended in 250 µL of YPD media (1% yeast extract (Oxoid,LP0021B), 2% peptone (Millipore,18332) and 2% glucose (Fischer UK, 10335850)) and incubated for 2 h at 30 °C for recovery. The entire contents of the microcentrifuge tube were then plated onto YPD + G418 (Sigma-Aldrich, A1720) at 200 mg/L and incubated for 48 h at 37 °C for colonies to form. The t*HMG1* cassette (*PTDH3-tHMG1-TCYC1*) integration was confirmed by PCR followed by Sanger sequencing to confirm correct in vivo assembly. The confirmed colonies were grown on YPD liquid media without selection to remove pCAS plasmid and streaked on YPD plates to obtain individual colonies. The plate was replicated onto YPD media with G418 to identify colonies that had lost pCAS plasmid and these were used for subsequent analysis.

## GC- and LC-MS analysis of yeast material

Three independent *S. cerevisiae* transformants, confirmed by colony PCR using pBEVY-L specific primers were grown at 30 °C and 210 rpm in Synthetic Complete (SC) selective medium lacking leucine (SC-LEU) in shake flasks for 72 h. For the analysis of peplusol content in whole cultures, 6 mL of culture was used for the extraction with an equal volume of ethyl acetate. Extraction was carried out overnight with vigorous shaking at room temperature. The ethyl acetate fraction was transferred to a glass tube and evaporated using a laboratory eva-porator (EZ-2 series, Genevac Ltd.). The final pellet was dissolved in 200 µL of methanol and used for LC-MS analysis as below. For the analysis of peplusol in media and cell pellets, 4 mL of culture was spun down at 3000 g for 5 min and 3 mL of the media supernatant was extracted with an equal volume of ethyl acetate, evaporated, re-suspended in 200 µL of methanol and used for LC-MS analysis as described above for the analysis of plant material. Cell pellets were mixed with 0.5 mL of methanol in 2 mL Eppendorf tubes, vortexed at 1,500 rpm at 60 °C for 10 min and spun down for 2 min at 20,000 g in a table-top microcentrifuge. 300 µL of the supernatant was run on LC-MS as described below. The cell pellet, whole culture and media-extract were each run on LC-MS using: (A) 10 mM ammo-nium formate in methanol/water 60:40 + 0.1 % formic acid, and (B) 10 mM ammonium formate in methanol/isopropanol 10:90 + 0.1 % formic acid as described in section "GC- and LC-MS analysis of plant material" above. A dilution series of *E. peplus* purified and NMR-verified peplusol standard was run in parallel to create a 7-point LC-MS stan-dard curve (0, 0.15625, 0.3125, 0.625, 1.25, 2.5 and 5 mg/mL). The standard curves (with linear regression $R^2 \geq 0.995$) were then used to calculate the amount of peplusol in the yeast extracts as presented in Figs. 3 and 6.

For squalene extraction, we modified an existing protocol[50] which involved spinning down 2 mL of a yeast cell culture for 2 min at 20,000 g in a table-top microcentrifuge. Resulting cell pellets were treated with 0.5 mL of saponification solution (25% EtOH, 20% KOH) for 2 h at 65 °C with intermittent agitation. 500 µL of hexane was added and samples were vortexed at 1,500 rpm for 10 min. 350 µL of the hexane upper phase was transferred into a glass vial and dried down in a laboratory evaporator (EZ-2 series, Genevac Ltd.). The dried extract was derivatized and analysed on GC-MS as described above for the analysis of plant material. Squalene was quantified against the external standard curve created from a commercial squalene standard (Sigma, cat. No. S3626) using extracted ion *m/z* 69. The squalene peak from yeast extracts had nearly identical *m/z* profile to the commercial squalene standard as shown in Supplementary Fig. 3C.

## Phylogenetic analysis

BLASTP searches were carried out against the non-redundant protein sequences database at the National Center for Biotechnology Infor-mation (NCBI, https://www.ncbi.nlm.nih.gov) for the seed plants, using the protein sequence of the *E. peplus* squalene synthase EpSS-L2 (WCJ40907.1) as a query. All 1397 sequences were retrieved with an expected value lower than 1e-100, including the *E. peplus* peplusol synthase EpSS-L1 (WCJ40906.1) which showed an expected value of 0.0. To extend coverage of squalene synthase homologues in the *Euphorbia* genus, TBLASTN searches were performed with the same query sequence against the transcriptome shotgun assembly database in this taxonomic domain as well as genome assemblies of *E. lathyris*, *E. pulcherrima* and *E. tirucalli*. Five full length and one near full length fragment were conceptually translated and annotated based on homology, and added to the above set along with the sequences of ElSS-L1 and ElSS-L2 (peplusol and squalene synthase of *E. lateriflora*).

We used online Kalign multiple sequence alignment software[51] (https://www.ebi.ac.uk/Tools/msa/kalign) to obtain protein sequence alignment. A subset of 500 sequences were obtained by removing sequences that contained large insertions-deletions and extra copies of near identical multiple entries at the genus level, guided by the Kalign alignment and the guide tree. For this smaller subset, we constructed a gene tree as follows: (1) we used MUSCLE v3.2 software[52] to build the alignment; (2) we generated conservative alignment blocks by removing highly polymorphic regions with the Gblocks v0.9144 software[53]; (3) we constructed the gene tree with Bayesian analyses using MrBayes v3.2.665[54]; and (4) the tree was visualised using Figtree v1.4.3 program (http://tree.bio.ed.ac.uk/software/figtree/). We repeated the above gene tree analysis with the 404 sequences from the monophyletic clade that are exclusively from the eudicots of the flowering plants (Supplementary Data 2), as the smaller data set provided more conserved residues in the alignment for gene tree construction. The edited tree with well-supported clades collapsed at taxonomic levels of orders and/or families is presented in Fig. 5.

## Protein modelling and structural analysis

cDNA-predicted amino acid sequences for EpSS-L1, EpSS-L2, ElSS-L1 and ElSS-L2 were used as input for the 3D structure prediction using the AlphaFold3 server (https://alphafoldserver.com/), with no ligands selected. The highest confidence model for each sequence was selected for further structural analysis using UCSF ChimeraX version 1.7 (2023-12-19). AlphaFold 3 modelled structures were overlaid with crystal structures of the human squalene synthase in complexes with magnesium ions and either presqualene pyropho-sphate (PDB ID: 3WEH) or farnesyl thiopyrophosphate (PDB ID; 3WEG). Both PDB files were used as a reference in the matchmaker module of UCSF ChimeraX software version 1.7[55], with the following parameters: a) chain pairing−best alignment pair of chains between reference and match structure; b) alignment−Needelman-Wunsh sequence alignment with BLOSUM62 matrix, gap opening penalty

−12 and gap extension penalty − 1; c) fitting with iteration by pruning long atom pairs and interaction cut-off distance of 2.

## Expression of recombinant proteins in *E. coli*

Alpha-fold 3 predicted models for AtSS, EpSS-L1, and EpSS-L2 were used to design protein truncations that would be expected to increase the probability of expressing soluble protein in *E. coli*. Ten amino acids of N-terminal unstructured sequence as well as C-terminal predicted transmembrane alpha-helices were removed. The remaining amino acid sequences for AtSS (10-377aa), EpSS-L1 (10-364aa), EpSS-L2 (10-375aa), EpSS-*L2to-L1* (10-375aa) and *EpSS-L1to-L2* (10-373aa) heptas-waps were codon-optimized for expression in *E. coli*. Gene fragments were synthesised as gBlock DNA fragments from IDT (Integrated DNA Technologies Inc.) with overhangs allowing direct insertion into a HindIII/KpnI digested pOPIN-F vector, downstream of a His$_6$-Tag (Addgene ID#26042), without PCR amplification, via In-Fusion cloning tools (TaKaRa bio Inc. Kusatsu, Japan), according to the manufacturer's protocol (sequences listed in Supplementary Data 1).

All five proteins were heterologously expressed in *E. coli* BL21(DE3)pLysS (Agilent Technologies, cat # 200132). Cells were transformed with pOPIN-F protein constructs as well as the empty vector (EV), according to the supplier's protocol, streaked for single colonies and inoculated into 5 mL of LB medium supplemented with 100 µg/mL carbenicillin and 50 µg/mL chloramphenicol. This seed culture was cultivated for 16–20 h at 37 °C and 220 rpm. An aliquot of the seed culture (diluted to final OD600 = 0.1) was inoculated into 2YT Medium (100 mL/500 mL) supplemented with 100 µg/mL carbenicillin and 50 µg/mL chloramphenicol. The resulting production culture was incubated at 37 °C and 200 rpm until the optical density (OD600) reached 0.4–0.6. Protein expression was induced by adding isopropyl-β-D-1-thiogalactopyranoside (IPTG) to a final concentration of 500 µM and cells were cultivated at 18 °C and 220 rpm for up to 24 h. The culture medium was removed by centrifugation (4 °C; 10 min; 3,200 g) and the cell pellets resuspended in 5 mL of lysis buffer (100 mM HEPES, pH7.5 or pH6.5). Resuspended cells were lysed using an ultrasonic liquid processor (Misonix® S-4000; 25 % amplitude; 2 s on/3 s off; total on-time: 4 min). Cell debris was removed by centrifugation (4 °C, 20 min, 35000 g) and clarified lysate was used in the enzymatic reactions.

## Enzyme reactions

Reaction mixtures (100 µL total volume) comprised of 50 mM HEPES at either pH 6.5 or 7.5 and various combinations of 0.5 mM FPP substrate, 30 mM co-factor (NADPH, NADP+) plus 25 mM MgCl$_2$ as detailed in Table 1. Reactions were initiated by addition of crude lysate (containing 220 µg of total protein) and incubation for 5 h at 30 °C with gentle agitation. Reaction products were extracted twice with 200 µL of hexane and twice with 200 µL of ethyl acetate. Organic solvents were evaporated in a GeneVac personal evaporator (Genevac, Ipswich, UK), residue derivatized using a mixture of pyridine (60 µL), N-methyl-N-(trimethylsilyl)trifluoroacetamide (MSTFA, 30 µL) and trimethylsilyl chloride (TMS, 1 µL) and analysed by GC-MS as described above. Peplusol and squalene standards were derivatized and run in parallel.

## Reporting summary

Further information on research design is available in the Nature Portfolio Reporting Summary linked to this article.

## Data availability

The *ElSS-L1* and *ElSS-L2* CDS sequence data generated in this study have been deposited in GenBank under accessions PP978604 and PP978605. *E. lateriflora* RNAseq data are available as GenBank Bio-Project ID PRJNA1131357. Mass spectrometry data are provided in Supplementary Fig. 3 and 9. Source data are provided with this paper.

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

## Acknowledgements

This research was funded by: the Garfield Weston Foundation, BBSRC and Innovate UK under grant number BB/M018210/01 and BBSRC Prosperity Partnership grant number BB/Y003217/1. B.R.L. and S.H.S. were supported by UKRI, grant number MR/S01862X/1. We thank Frank Van Caekenberghe (Botanic Garden Meise, Belgium) for sharing the *E. lateriflora* specimen GH-0-BR-2014067883 and horticultural advice on growing the plant material. We thank Prof. George Lomonossoff (John Innes Centre, Norwich) for providing the pEAQ-HT vector. We thank Dr Sally James and Dr Tony Larson from the Bioscience Technology Facility at the University of York for assistance with RNAseq of *E. lateriflora* and metabolomics respectively, Mr Gianluca Ruvo at Rothamsted Research Metabolomics Facility for the collection of NMR data, Dr Cobus Smit and Dr Catharine Wood for advice on in vitro expression of recombinant proteins in *E. coli* and Mr George Brown for graphical assistance with figure preparation.

## Author contributions

T.C. and I.A.G. designed research; T.C., Y.L.,A.D.G., D.H., S.H.S., and J.L.W. performed research; T.C., Y.L., B.R.L., J.L.W. and I.A.G. analysed data; T.C., Y.L., B.R.L., and I.A.G. wrote the paper (equal contribution for T.C. and I.A.G., supporting role for Y.L. and B.R.L).

## Competing interests

The authors declare no competing interests.
