## [Peer Review file · Nature Communications]

Evolution of linear triterpenoid biosynthesis within the *Euphorbia* plant genus

Corresponding Author: Professor Ian Graham

Version 0:

Reviewer comments:

Reviewer #1

(Remarks to the Author)

The manuscript by Czechowski et al on the 'Evolution of linear triterpenoid biosynthesis within the *Euphorbia* plant genus' reports on the identification of the enzyme responsible for the biogenesis of the acyclic triterpene alcohol peplusol found in two species of the euphorbia genus *E. peplus* and *E. lateriflora*. The approach for the isolation of that enzyme is based on a hypothetical mechanism for peplusol formation that would imply a head-to-head condensation of farnesyl diphosphates and formation of peplusol diphosphate and not pre-squalene diphosphate en route to squalene. Paralogues of the well-known squalene synthase gene identified as squalene synthase-like genes/enzymes (SS-Like) are unveiled here, and functionally characterized as peplusol synthases from *E. peplus* genome and then *E. lateriflora* transcriptome. It is found that SS-L1 expressed in the latex of *E. peplusol* makes peplusol upon ad hoc expression in *Nicotiana benthamiana* agroinfiltrated leaves and in transformed yeast. It is also shown that SS-L2 which displays an ubiquitous expression profile in organs is a squalene synthase.

The authors took an elegant approach of enzyme swapping after modelling the SS-Ls structures (alpha fold 3) with a reference to available crystal structures of human squalene synthase and were able to generate hexaswaps of SS-L1 with squalene synthase activity and vice versa hexaswaps of SS-L2 with peplusol synthase activity, which is remarkable. This suggests that peplusol-synthase activity most likely evolved from squalene-synthase by gene duplication/neofunctionalization.

This is another example of a newly discovered biosynthetic route to a triterpene that do not require the otherwise mandatory squalene intermediate (Hui Tao et al 2022 Discovery of non-squalene triterpenes (<https://doi.org/10.1038/s41586-022-04773-3>)).

Going into the analytical work I have a number of questions for the authors.

Fig.2A. The compound (2) identified in Extracted Ion Chromatograms (EIC) at m/z 427.39 is not identified, could it be the C₃₀H₅₀O squalene oxide (2,3-oxidosqualene) produced by endogenous squalene epoxidase (from *N. benthamiana*) and serving as precursor for other triterpenes ?

In Fig.2A bottom chromatogram for HMGRt / EpSS-L2 is the peak (2) more intense than for EpSS-L2 alone ? If this is verified it could be that squalene produced in *N. benthamiana* leaves is epoxidized into 2,3-oxidosqualene. It is also possible that most of the engineered squalene is entering the post-squalene sterol pathway, then the level of total sterols is increased (the authors may check Bouvier-Navé et al 2010 Plant Physiol 152(1):107-19. doi: 10.1104/pp.109.145672).

The description of some parts of the analytical methods is confusing and possibly casts doubt on certain conclusions, specially for squalene analysis.

Line 430-448. Squalene analysis is performed on the unsaponifiable hexane extract of leaves or yeasts. I do not understand why the dry extract is derivatized with N-Methyl-N-(trimethylsilyl)TriFluoroAcetamide (MSTFA) for GC-MS analysis. This type of derivatization is used to produce TMS of triterpene alcohols for instance. MSTFA is used to change a H atom to a trimethylsilyl group in compounds bearing hydroxy, carboxy functions, but to the best of my knowledge squalene is unaffected by MSTFA in conditions described by the authors. What is the mass spectra of derivatized squalene ?

Did the authors check in their GC-MS runs if derivatives of peplusol like TMS-peplusol is visible ? I am surprised that this is not described here since peplusol was isolated from the latex of *E. peplus* to get a standard compound, which could have been used in GC as well (then peplusol, squalene and other triterpenes are detectable in a same extract).

Is it possible that peplusol produced in *N. benthamiana* is taken by some conjugating enzyme to produce derivatives (e.g. esters) that would escape detection by LC-MS because in that LC-MS assay the extract is an ethyl acetate extract ? In a saponified extract, peplusol found as a conjugate would be released (i.e. are peplusol esters or glucosides found in SS-L1 infiltrated *N. benthamiana* leaves ?).

Other points.

Line 261. Mentions to Fig.6A and Fig.6B are not indicated in the text

Typos please check throughout , some examples here:

Line 437. Derivatised.

Line 769. Predicted

Fig.S3A is difficult to read.

Materials Methods

Line 415 says "Analytical methods for squalene and peplusol extraction, detection and quantification were optimised to obtain results presented in Fig. 1, Fig. 3 and Fig. S5", however there are no analytical data in Fig.1. Please check.

Line 428 says "The standard curve (with linear regression $R^2 \geq 0.998$) was used to calculate the amount of peplusol in the plant extracts as presented in Fig. 2 and Fig. 4". There are peplusol quantification in 2B, 3A, 4A.

Line 446-448 squalene quantification is shown in Fig.2, 3, 4, 6 not only 2 and 4.

Line 494 to 512 is redundant (describes the saponification procedure and hexane extraction as in line 430 to 448).

Reviewer #2

(Remarks to the Author)

This is a review of the article entitled "Evolution of linear triterpenoid biosynthesis within the *Euphorbia* plant genus" by Czechowski et al. The manuscript aims to uncover the evolutionary origins and biochemical functions of peplusol synthases in the *Euphorbia* genus, exploring how these enzymes differ from squalene synthases. Figure 1 presents the structures of key linear triterpenes and the biosynthetic pathways for squalene and peplusol. Figures 2-4 show functional characterization of squalene synthase-likes in *N. benthamiana* and in *Saccharomyces cerevisiae*. Figure 5 shows a phylogenetic analysis of squalene synthase likes across many dicots, and Figure 6 highlights the structural analysis and site-directed mutagenesis, identifying critical residues that influence peplusol versus squalene synthase activity. Overall, the figures are of high quality, the data are sound, and the major conclusions and findings are supported by the data. The scope of the work is appropriate for *Nature Communications*. I have a few comments on how the manuscript could be improved, detailed below.

Major comments:

- The authors test two combinations of residues for site directed mutagenesis and find that each set is important for peplusol synthesis activity. In the reciprocal engineering (trying to make a squalene synthase make peplusol), were all five residues tried simultaneously? Perhaps all five residues together could make a squalene synthase into a peplusol synthase.

Minor comments:

- The final sentence of the abstract "[our discovery] places peplusol in a unique group of C30 compounds". I think that peplusol naturally belongs to this group, regardless of any research that could be done. Can a more precise wording be selected to indicate that this work reveals peplusol's biosynthesis rather than categorically placing it within this group?

- "both highly expressed in latex, a peplusol rich tissue". Is latex really a tissue? I think most people consider it to be a milky, liquid substance. Perhaps the wording can be improved here?

- abstract: "active site transplantations": I feel like these experiments would be best described as site directed mutagenesis, since "transplantation" suggests something more along the lines of domain swapping.

- It is interesting that the authors note active site cavity size as well as shape differences between the two types of synthases. I appreciate this and hope to see further articles in which not only amino acid residues, but also active site shapes, are considered.

Reviewer #3

(Remarks to the Author)

The manuscript by Czechowski et al. reports on the discovery of peplusol synthases from *Euphorbia* species. Peplusol is a unique linear triterpenoid that could have interesting applications. The authors discuss the evolution and mechanism of these enzymes and use heterologous hosts, such as tobacco plants and yeast, to characterize their activity in vivo. By identifying key residues in the protein structure of peplusol synthase, the authors demonstrate the conversions of this enzyme into a squalene synthase and the opposite. While this research is novel and timely, more data are needed to support these findings.

Major comments:

The activity of new enzymes and mutants characterized in vivo must be demonstrated by in vitro enzymatic assays, particularly because of the endogenous squalene produced in the host organisms used in this study.

In Fig. 2A, the unknown compound 2 produced in *N. benthamiana* must be identified by NMR analysis. It is produced by EpSS-L1 at elevated levels of FPP due to tHMGR overexpression. Please show the chromatograms from yeast samples, as well, and provide the mass spectra of standards and products from both *N. benthamiana* and yeast. Please include squalene standard as a control.

Line 148-150: How do you explain the elevation of squalene by overexpression of EpSS-L1 in yeast?

Line 158: Where do you show the 220- and 540-fold increase in the squalene level in yeast by overexpression of EpSS-L2 and AtSS.

Line 271: Fig. 6D does not show a significant decrease of squalene production by the EpSS-L2to-L1 hexaswap compared to the EpSS-L2.

In Materials and Methods, please separate the cloning from GC and LC analysis.

Line 446-447: There is no need to specifically derivatize squalene because the TMS derivatization method displaces the active hydrogen atoms of hydroxyl and carboxyl groups (which are non-existent in squalene) to form silyl groups. Squalene can be detected by GC without derivatization. Peplusol, on the other hand, can be derivatized due to its OH group and can likely be detected by GC. I expect that, with derivatization, both peplusol and squalene can be detected in the same sample, which is essential to support these findings.

Minor comments:

Line 157: There is a parenthesis typo after AtSS.

Line 161: Please check and apply yeast nomenclature for the yeast genes and proteins. ERG9 is the gene. You are mentioning the squalene synthase, which should be labeled Erg9p.

In Fig. 5 the shaded rectangles used to highlight the sister clades of *Euphorbia* peplusol and squalene synthases hinder proper reading of the figure. I recommend using a lighter color and displaying these rectangles below the clades rather than above.

Reviewer #4

(Remarks to the Author)

General comments: The manuscript under review offers a focused exploration on uncovering a neofunctionalized peplusol synthase that directly produces peplusol instead of squalene, thus placing peplusol in a distinct category of C30 compounds not derived from squalene. The identification of key residues determining product profiles is a significant contribution. The finding that squalene synthase can be converted to peplusol synthase and vice versa through active site transplantations is truly remarkable. Additionally, the demonstration of the feasibility of the evolution of peplusol synthase from the ancient squalene synthase as suggested by phylogenetic analysis is an important achievement. Overall, this study provides fresh insights into the biosynthesis of plant natural products and holds great potential for the development of novel bioactive compounds and production strategies.

However, in its current form, the manuscript requires further revisions for publication in *Nature Communication*. Some experimental conclusions appear less than fully rigorous. It would be beneficial for the authors to provide more comprehensive data or alternative explanations to strengthen these conclusions. Moreover, it is necessary to consider adding necessary relevant evidence. Additionally, the article's structure could be improved to enhance the flow of the narrative, and the language could be made clearer to facilitate understanding for a wider audience.

The following recommendations and suggestions as following,

Major comments:

Page 4

Line 130-132

"Transient expression of EpSS-L1 in combination with HMGRt significantly reduced the level of squalene (12-fold) in *N. benthamiana* leaves, suggesting that the introduced EpSS-L1 is competing for FPP with the endogenous *N. benthamiana* squalene synthase (Fig. 2C)."

The conclusion here points out whether there is direct evidence that the introduction of EpSS-L1 competes with squalene synthase for FPP. There is no characterization result of FPP content in the full text. Relevant FPP content data can be supplemented to provide direct evidence. If this is difficult, please revise the relevant conclusion to a more rigorous one.

Page 5

In the first part of the results and discussion, there is a lack of detailed verification of key experimental steps. For example, when determining the functions of EpSS-L1 and EpSS-L2, only inference is made through expression in *Nicotiana benthamiana* and *Saccharomyces cerevisiae*, and there is no further direct evidence to confirm their mechanism of action.

Page 6

Line 187-190

"Given the high sequence homology with EpSSL-2 which predominantly exhibits squalene synthase activity (Fig 3) we suggest that EISS-L2 likely also encodes a squalene synthase."

ELSS-L2 did not increase squalene production compared to the control. How can it be defined as a squalene synthase?

Page 6

The study is mainly focused on a few *Euphorbia* species, which may not be representative of the entire genus. A larger number of species could be included to strengthen the generality of the findings.

Page 6-8

The manuscript infers the evolutionary origin of peplusol synthase relying on phylogenetic gene tree analysis and active site transplantation experiments. It will be better to combine other evolutionary analysis methods for more comprehensive understanding of the evolutionary relationship between peplusol synthase and squalene synthase?

Minor comments:

Page 3

Line 83-91

When discussing the evolution of squalene synthase genes, it will be better to combine the discovery of peplusol synthase more closely?

Page 5

Line 155

The Squalene Synthase from *A. thaliana* has been defined as AtSS earlier. Thus, the abbreviation can be used in the subsequent text. Similar issues throughout the full text should also be corrected.

Version 1:

Reviewer comments:

Reviewer #1

(Remarks to the Author)

The manuscript by Czechowski et al on the 'Evolution of linear triterpenoid biosynthesis within the *Euphorbia* plant genus' has been very carefully and accurately revised in response to the questions I had. The experimental revisions and the clear explanations and clarifications on the analytical methods are completely convincing.

Reviewer #2

(Remarks to the Author)

The authors have addressed all of my comments.

Reviewer #3

(Remarks to the Author)

The authors have successfully addressed my concerns, and the manuscript is now substantially improved and suitable for publication. New experimental evidence has been provided for the detection and quantification of the major compounds produced in this study, including NMR data confirming the production of 2,3-oxidosqualene. The *in vitro* characterization of peplusol synthase was performed using crude protein lysates. While purified proteins would be preferred for increased confidence in the results, obtaining these fractions can be highly challenging and varies considerably from protein to protein. Importantly, the results obtained with crude lysates are consistent with the main findings observed in the *in vivo* systems, with only a few discrepancies. These discrepancies may be attributed to the need for specific enzymatic assay conditions, differences in expression systems (*E. coli*, *S. cerevisiae*, and *N. benthamiana*), or the absence of a membrane environment due to protein truncations.

I have only a minor suggestion regarding the visibility of the chromatograms in Figures 3A and S7, which are currently difficult to read. I understand that some software may produce low-quality chromatograms or that more thorough processing may be required to achieve publication-quality resolution. I am confident that the authors can find a solution to improve the quality of these figures.

Reviewer #4

(Remarks to the Author)

The authors have addressed all my concerns. I have no more questions.

RESPONSE TO REVIEWER COMMENTS

Reviewer comments in Arial 12 point black font

Author responses in Arial 12 point red font

Amended text from revised manuscript shown in Arial 10 point black font

Reviewer #1 (Remarks to the Author):

The manuscript by Czechowski et al on the ‘Evolution of linear triterpenoid biosynthesis within the Euphorbia plant genus’ reports on the identification of the enzyme responsible for the biogenesis of the acyclic triterpene alcohol peplusol found in two species of the euphorbia genus E. peplus and E. lateriflora. The approach for the isolation of that enzyme is based on a hypothetical mechanism for peplusol formation that would imply a head-to-head condensation of farnesyl diphosphates and formation of peplusol diphosphate and not pre-squalene diphosphate en route to squalene. Paralogues of the well-known squalene synthase gene identified as squalene synthase-like genes/enzymes (SS-Like) are unveiled here, and functionally characterized as peplusol synthases from E. peplus genome and then E. lateriflora transcriptome. It is found that SS-L1 expressed in the latex of E. peplusol makes peplusol upon ad hoc expression in Nicotiana benthamiana agroinfiltrated leaves and in transformed yeast. It is also shown that SS-L2 which displays an ubiquitous expression profile in organs is a squalene synthase.

The authors took an elegant approach of enzyme swapping after modelling the SS-Ls structures (alpha fold 3) with a reference to available crystal structures of human squalene synthase and were able to generate hexaswaps of SS-L1 with squalene synthase activity and vice versa hexaswaps of SS-L2 with peplusol synthase activity, which is remarkable. This suggests that peplusol-synthase activity most likely evolved from squalene-synthase by gene duplication/neofunctionalization.

This is another example of a newly discovered biosynthetic route to a triterpene that do not require the otherwise mandatory squalene intermediate (Hui Tao et al 2022 Discovery of non-squalene triterpenes (<https://doi.org/10.1038/s41586-022-04773-3>)).

Thanks to reviewer 1 for the positive comments on the importance of our findings and the quality of our experiments. In the revised manuscript we now cite the Hui Tao et al. 2022 paper at the end of the conclusions section (page 11 lines 428-430):

“Our findings expand the knowledge, of very rare non-squalene-dependent triterpene biosynthesis, beyond the recently discovered pathways operating in fungi. (Tao et al 2022)”

Going into the analytical work I have a number of questions for the authors.

Fig.2A. The compound (2) identified in Extracted Ion Chromatograms (EIC) at m/z 427.39 is not identified, could it be the $C_{30}H_{50}O$ squalene oxide (2,3-oxidosqualene) produced by endogenous squalene epoxidase (from *N. benthamiana*) and serving as precursor for other triterpenes ?

To address this question we have now extracted and purified compound 2 from infiltrated leaves of *N. benthamiana*, performed NMR characterisation and confirmed its identity as 2,3-oxidosqualene.

We have altered the text, lines 124 - 128 page 4 to read:

“Compound (2) was extracted from infiltrated leaves of *N. benthamiana*, purified and characterized by NMR, revealing its identity as 2,3-oxidosqualene. This compound can be formed by oxidation of squalene by squalene monooxygenase^{31,32} We assume 2,3-oxidosqualene is produced by an endogenous squalene monooxygenase in *N. benthamiana* to generate the substrate for various oxidosqualene cyclases involved in essential sterol biosynthesis.“

In Fig.2A bottom chromatogram for HMGRt / EpSS-L2 is the peak (2) more intense than for EpSS-L2 alone ? If this is verified it could be that squalene produced in N.benthamiana leaves is epoxidized into 2,3-oxidosqualene. It is also possible that most of the engineered squalene is entering the post-squalene sterol pathway, then the level of total sterols is increased (the authors may check Bouvier-Navé et al 2010 Plant Physiol 152(1):107-19. doi: 10.1104/pp.109.145672).

Yes, this peak is more intense when *EpSS-L2* is co-expressed with *HMGRt*. It is also possible that the transient expression of *tHMGR* in *N. benthamiana* elevates the level of total sterols.

We have now provided quantitative data on the levels of 2,3-oxidosqualene in *N. benthamiana* as detailed in the following text:

Page 4, lines 136-143:

“Levels of 2,3-oxidosqualene were also significantly reduced when *EpSS-L1* was transiently expressed with (5-fold) and without (10-fold) *HMGRt* (Fig 2D and E)., These results may be due to suggesting that the introduced *EpSS-L1* is competing with the endogenous *N. benthamiana* SQUALENE SYNTHASE for FPP precursor but this would need to be confirmed by assaying FPP levels. There is no significant change in squalene levels when *AtSS* or *EpSS-L2* are expressed transiently with *HMGRt*, (Fig. 2C). However levels of 2,3-oxidosqualene significantly increased when *AtSS* and *EpSS-L2* are expressed without *HMGRt* (4-fold and 6.7-fold, respectively, Fig 2D), which indicates *EpSS-L2* encodes a functional squalene synthase”.

The description of some parts of the analytical methods is confusing and possibly casts doubt on certain conclusions, specially for squalene analysis. Line 430-448. Squalene analysis is performed on the unsaponifiable hexane extract of leaves or yeasts. I do not understand why the dry extract is derivatized with N-Methyl-N-(trimethylsilyl)TriFluoroAcetamide (MSTFA) for GC-MS analysis. This type of derivatization is used to produce TMS of triterpene alcohols for instance. MSTFA is used to change a H atom to a trimethylsilyl group in compounds bearing hydroxy, carboxy functions, but to the best of my knowledge squalene is unaffected by MSTFA

in conditions described by the authors. What is the mass spectra of derivatized squalene ?

Did the authors check in their GC-MS runs if derivatives of peplusol like TMS-peplusol is visible ? I am surprised that this is not described here since peplusol was isolated from the latex of E.peplus to get a standard compound, which could have been used in GC as well (then peplusol, squalene and other triterpenes are detectable in a same extract).

Is it possible that peplusol produced in N.benthamiana is taken by some conjugating enzyme to produce derivatives (e.g. esters) that would escape detection by LC-MS because in that LC-MS assay the extract is an ethyl acetate extract ? In a saponified extract, peplusol found as a conjugate would be released (i.e. are peplusol esters or glucosides found in SS-L1 infiltrated N.benthamiana leaves ?).

We agree with the reviewer that referring to squalene as being “derivatized” in the text is confusing; more correctly the extracts were subjected to a silylation procedure where some components would be derivatized, but not squalene. We have made edits to the text to clarify this (see line 577-578 and 640). We performed silylation to improve the chromatographic peaks shapes and elution profiles of components in the extracts that could otherwise co-elute and confound squalene measurements.

The GC-MS quantification method for squalene in yeast- and plant-derived samples was based on a published method to quantify a range of hydroxylated compounds by GC-MS as well as squalene. We have indeed observed that TMS-peplusol was visible in TMS-derivatized yeast- and plant-derived samples. Mass spectra of TMS-derivatized peplusol (example1 below) shows m/z 69 as the most abundant ion and m/z 499 as the highest molecular weight ion observed, the latter corresponding to the mass of the intact peplusol molecule (m/z 426) plus a TMS-group (m/z 73).

Derivatized peplusol peak mass spectra

D13_11_24_PO_5_1 #21478 RT: 22.90 AV: 1 NL: 5.10E6
 F: + c EI Full ms [50.00-700.00]

D13_11_24_D_YTC2_7_1 #21427 RT: 22.86 AV: 1 NL: 1.06E6
 F: + c EI Full ms [50.00-700.00]

Example 1

Peplusol (unlike squalene) is detectable by GC-MS only after TMS derivatization as can be seen from the comparison of MSTFA treated vs non-treated extracts from yeast strain overexpressing EpSS-L1 in the tHMG1 background (example 2).

RT: 21.00 - 24.00

RT: 21.00 - 24.00

Example 2

It is however very evident that the GC-MS method we have used to quantify squalene cannot be used for the detection and quantification of peplusol in *N. benthamiana* plant-derived samples (example 3). Although the method provides enough resolution to separate sterols derived from yeast extracts, there are broad coeluting peaks in plant-derived samples, especially evident in the “high FPP” background, when truncated HMGR is overexpressed together with peplusol synthase. The GC-MS method therefore does not allow us to compare peplusol levels between yeast and plant-derived samples; this is the key reason we used LC-MS to quantify peplusol across all expression systems.

Example 3

We cannot exclude the possibility that peplusol produced in *N. benthamiana* could be transformed by a conjugating enzyme to produce derivatives (e.g. sugar esters), that might escape detection by LC-MS. Nevertheless, by detecting peplusol in *N. benthamiana* leaves that are transiently expressing the candidate gene we believe we have demonstrated triterpenoid biosynthesis, specifically peplusol synthase activity, which was the main point of the study. Additionally, our method developed for LC-MS separates peplusol from other related C30 compounds very well as evident from Figure 2 in the submitted manuscript. The other advantage of the LC-MS method we have used for peplusol is that the combination of soft ionization and high resolution MS produces an ion that allows the assignment of molecular formulae. This isn't possible with GC-MS, which relies on fragment similarity to related compounds or library matches to shortlisted candidates. Finally, peplusol analysis as TMS-

derivatives by GC-MS adds extra sample preparation steps that could lead to greater analytical variability than the simpler LC-MS method we have used.

To further reassure Reviewer 1 about the effect of the TMS-derivatization on squalene quantification we have performed the following experiment. We grew three independent *S. cerevisiae* clones transformed with either Empty Vector or EpSS-L1 (renamed as EpPS in this manuscript) in the tHMG1-expressing background and performed saponification and hexane-extraction, following the protocols described in the materials and methods. We split the hexane extract into two equal aliquots - one was derivatized using MSTFA and TMS and the other was left non-derivatized. We also produced two squalene standard curves with identical concentrations of squalene, one having been TMS-derivatized and the other not derivatized as shown in example 4 below.

Example 4

Example 5 below provides quantification of squalene in TMS-derivatized yeast samples based on TMS-derivatized squalene standard curve and in non-derivatized yeast samples based on non-derivatized squalene standard curve. It is quite evident that the final squalene concentrations (mg/L) measured are indeed very similar between TMS-derivatized and non-derivatized yeast samples. There is no statistically significant difference between derivatized vs non-derivatized sample sets and the small discrepancies observed between two sample sets are well within biological / technical variation observed for the experiment.

Genotype	Clone	Derivatised	Sample	PeakArea (m/z 69)	ng/inj	mg/L
+tHMG1-EV	5	no	ND_YTC5_5	17421212	188.9666	31.49443914
+tHMG1-EV	6	no	ND_YTC5_6	13898388	150.7548	25.12580267
+tHMG1-EV	7	no	ND_YTC5_7	17587024	190.7651	31.79419762
+tHMG1-EpSS-L1	5	no	ND_YTC2_5	29471666	319.6770	53.27950726
+tHMG1-EpSS-L1	6	no	ND_YTC2_6	17989079	195.1262	32.52104123
+tHMG1-EpSS-L1	7	no	ND_YTC2_7	25640203	278.1174	46.35290662
+tHMG1-EV	5	yes	D_YTC5_5	15738903	205.2891	34.21485776
+tHMG1-EV	6	yes	D_YTC5_6	12445661	162.3340	27.05566715
+tHMG1-EV	7	yes	D_YTC5_7	17266109	225.2091	37.53485637
+tHMG1-EpSS-L1	5	yes	D_YTC2_5	22469117	293.0741	48.84569415
+tHMG1-EpSS-L1	6	yes	D_YTC2_6	12893838	168.1797	28.02996074
+tHMG1-EpSS-L1	7	yes	D_YTC2_7	22929697	299.0817	49.84695067

Example 5

Other points.

Line 261. Mentions to Fig.6A and Fig.6B are not indicated in the text

Corrected

Typos please check throughout , some examples here:

Line 437. Derivatised.

Line 769. Predicted

Corrected

Fig.S3A is difficult to read.

Size of Fig.S3A has been increased and renamed to Fig.S4A

Materials Methods

Line 415 says “Analytical methods for squalene and peplusol extraction, detection and quantification were optimised to obtain results presented in Fig. 1, Fig. 3 and Fig. S5”, however there are no analytical data in Fig.1. Please check.

Thanks for pointing out this error, we were actually referring to data in Figure 2. This has now been corrected.

Line 428 says “The standard curve (with linear regression $R^2 \geq 0.998$) was used to calculate the amount of peplusol in the plant extracts as presented in Fig. 2 and Fig. 4”. There are peplusol quantification in 2B, 3A, 4A.

Thank you for pointing this out. For clarity we have modified the Materials and Methods so that there are now two separate sections for GC- and LC-MS based analysis of plant (page 15) and yeast (page 17) material. The section on plant material details squalene and peplusol quantification presented in Fig. 2 and Fig. 4 and the section on yeast material details squalene and peplusol quantification presented in Fig. 3 and Fig. 6

Line 446-448 squalene quantification is shown in Fig.2, 3, 4, 6 not only 2 and 4.

Thank you, the response to the previous comment refers.

Line 494 to 512 is redundant (describes the saponification procedure and hexane extraction as in line 430 to 448).

The text has been modified to reduce the redundancy referred to.

Reviewer #2 (Remarks to the Author):

This is a review of the article entitled "Evolution of linear triterpenoid biosynthesis within the Euphorbia plant genus" by Czechowski et al. The manuscript aims to uncover the evolutionary origins and biochemical functions of peplusol synthases in the Euphorbia genus, exploring how these enzymes differ from squalene synthases. Figure 1 presents the structures of key linear triterpenes and the biosynthetic pathways for squalene and peplusol. Figures 2-4 show functional characterization of squalene synthase-likes in N. benthamiana and in Saccharomyces cerevisiae. Figure 5 shows a phylogenetic analysis of squalene synthase likes across many dicots, and Figure 6 highlights the structural analysis and site-directed mutagenesis, identifying critical residues that influence peplusol versus squalene synthase activity. Overall, the figures are of high quality, the data are sound, and the major conclusions and findings are supported by the data. The scope of the work is appropriate for Nature Communications. I have a few comments on how the manuscript could be improved, detailed below.

Thanks to reviewer 2 for the very positive remarks on the manuscript.

Major comments:

- The authors test two combinations of residues for site directed mutagenesis and find that each set is important for peplusol synthesis activity. In the reciprocal engineering (trying to make a squalene synthase make peplusol), were all five residues tried simultaneously? Perhaps all five residues together could make a squalene synthase into a peplusol synthase.

To recap, we initially identified six consistently different amino acid residues around the catalytic sites of proteins exhibiting either squalene synthase or peplusol synthase activities.

Our experimental design swapped all six of these residues in both directions - i.e. we introduced the six residues from the peplusol synthase protein into the squalene synthase protein resulting in peplusol synthase activity being exhibited by the mutated squalene synthase protein. Swapping six squalene synthase residues into the peplusol synthase protein did not introduce squalene synthase activity (Figure 6).

Based on protein modelling we identified an additional candidate residue around the catalytic site that was consistently different between the two enzymes and when this was included with the other six residues as a hepta-swap the mutated peplusol synthase then exhibited squalene synthase activity as shown in Figure 6.

We did not try any further combinations (including swapping 5 residues) as we consider the results of the hexa and hepta-swaps sufficient to demonstrate the principle that the peplusol synthase has evolved from squalene synthase due to gene duplication and neo-functionalisation. Furthermore, adding more swap combinations would have exceeded the resource capacity available for the project.

Minor comments:

- The final sentence of the abstract "[our discovery] places peplusol in a unique group of C30 compounds". I think that peplusol naturally belongs to this group, regardless of

any research that could be done. Can a more precise wording be selected to indicate that this work reveals peplusol's biosynthesis rather than categorically placing it within this group?

Thanks for this comment.

In response, the final sentence of the abstract has been modified to read:

"Our discovery of a neofunctionalized peplusol synthase, that directly produces peplusol rather than squalene reveals the biosynthetic origin of another member of this rare group of linear triterpenes not derived from squalene".

- *"both highly expressed in latex, a peplusol rich tissue". Is latex really a tissue? I think most people consider it to be a milky, liquid substance. Perhaps the wording can be improved here?*

Thanks for this comment. The text referred to (line 109, Page 4) has been modified to read:

"revealed that they are both highly expressed in latex, - a milky sap secreted by laticiferous tissues."

- *abstract: "active site transplantations": I feel like these experiments would be best described as site directed mutagenesis, since "transplantation" suggests something more along the lines of domain swapping.*

Thanks for this comment.

"active site amino acid swapping" is now used instead of "active site transplantation"

- *It is interesting that the authors note active site cavity size as well as shape differences between the two types of synthases. I appreciate this and hope to see further articles in which not only amino acid residues, but also active site shapes, are considered.*

Thank you for this encouraging comment.

Reviewer #3 (Remarks to the Author):

The manuscript by Czechowski et al. reports on the discovery of peplusol synthases from Euforbia species. Peplusol is a unique linear triterpenoid that could have interesting applications. The authors discuss the evolution and mechanism of these enzymes and use heterologous hosts, such as tobacco plants and yeast, to characterize their activity in vivo. By identifying key residues in the protein structure of peplusol synthase, the authors demonstrate the conversions of this enzyme into a squalene synthase and the opposite. While this research is novel and timely, more data are needed to support these findings.

Thanks to reviewer 3 for the supportive comments on our manuscript.

Major comments:

The activity of new enzymes and mutants characterized in vivo must be demonstrated by in vitro enzymatic assays, particularly because of the endogenous squalene produced in the host organisms used in this study.

We have now conducted *in vitro* assays on the new enzymes and mutants as detailed on pages 9 - 12 of the revised manuscript with results in Table 1, Fig. S7 and S8. These largely confirm the in-vivo results and address the issue of endogenous squalene being produced in the host organisms.

In Fig. 2A, the unknown compound 2 produced in N. benthamiana must be identified by NMR analysis. It is produced by EpSS-L1 at elevated levels of FPP due to tHMGR overexpression.

This same point was raised by reviewer 1. As noted above, we have now extracted and purified compound 2 from infiltrated leaves of *N. benthamiana*, performed NMR characterisation and confirmed its identity as 2,3-oxidosqualene.

Please show the chromatograms from yeast samples, as well, and provide the mass spectra of standards and products from both N. benthamiana and yeast. Please include squalene standard as a control.

Figure 3 has now been modified to include chromatograms from yeast samples. Mass spectra of standards and products from both *N. benthamiana* and yeast are now included in new supplementary figure S2A-C.

Line 148-150: How do you explain the elevation of squalene by overexpression of EpSS-L1 in yeast?

The elevation of squalene by overexpression of EpSS-L1 in yeast can be explained by the fact that EpSS-L1 carries some additional squalene synthase activity as demonstrated by *in vitro* activity assays of the enzyme using *E. coli* extracts (Table 1).

Line 158: Where do you show the 220- and 540-fold increase in the squalene level in yeast by overexpression of EpSS-L2 and AtSS.

The fold increase in squalene that is referred to is shown in Figure 3B.

Line 271: Fig. 6D does not show a significant decrease of squalene production by the EpSS-L2to-L1 hexaswap compared to the EpSS-L2.

Statistical analysis of data shown in Fig. 6D has been extended to include the *EpSS-L2to-L1* both hexaswap and heptaswap. Both are significantly reduced compared to *EpSS-L2*.

In Materials and Methods, please separate the cloning from GC and LC analysis.

Materials and Methods have been modified such that the cloning has been separated for both plant- and yeast analysis.

Line 446-447: There is no need to specifically derivatize squalene because the TMS derivatization method displaces the active hydrogen atoms of hydroxyl and carboxyl groups (which are non-existent in squalene) to form silyl groups. Squalene can be detected by GC without derivatization. Peplusol, on the other hand, can be derivatized due to its OH group and can likely be detected by GC. I expect that, with derivatization, both peplusol and squalene can be detected in the same sample, which is essential to support these findings.

This issue was also raised by reviewer 1. Please see the detailed response to reviewer 1 above.

Minor comments:

Line 157: There is a parenthesis typo after AtSS.

Corrected

Line 161: Please check and apply yeast nomenclature for the yeast genes and proteins. ERG9 is the gene. You are mentioning the squalene synthase, which should be labeled Erg9p.

Corrected

In Fig. 5 the shaded rectangles used to highlight the sister clades of Euphorbia peplusol and squalene synthases hinder proper reading of the figure. I recommend using a lighter color and displaying these rectangles below the clades rather than above.

Figure 5 has been modified as requested.

Reviewer #4 (Remarks to the Author):

General comments: The manuscript under review offers a focused exploration on uncovering a neofunctionalized peplusol synthase that directly produces peplusol instead of squalene, thus placing peplusol in a distinct category of C30 compounds not derived from squalene. The identification of key residues determining product profiles is a significant contribution. The finding that squalene synthase can be converted to peplusol synthase and vice versa through active site transplantations is truly remarkable. Additionally, the demonstration of the feasibility of the evolution of peplusol synthase from the ancient squalene synthase as suggested by phylogenetic analysis is an important achievement. Overall, this study provides fresh insights into the biosynthesis of plant natural products and holds great potential for the development of novel bioactive compounds and production strategies.

Thanks to reviewer 4 for the supportive comments on our work.

However, in its current form, the manuscript requires further revisions for publication in Nature Communication. Some experimental conclusions appear less than fully rigorous. It would be beneficial for the authors to provide more comprehensive data or alternative explanations to strengthen these conclusions. Moreover, it is necessary to consider adding necessary relevant evidence. Additionally, the article's structure could

be improved to enhance the flow of the narrative, and the language could be made clearer to facilitate understanding for a wider audience.

The following recommendations and suggestions as following,

Major comments:

Page 4 Line 130-132

“Transient expression of EpSS-L1 in combination with HMGRt significantly reduced the level of squalene (12-fold) in *N. benthamiana* leaves, suggesting that the introduced EpSS-L1 is competing for FPP with the endogenous *N. benthamiana* squalene synthase (Fig. 2C).”

The conclusion here points out whether there is direct evidence that the introduction of EpSS-L1 competes with squalene synthase for FPP. There is no characterization result of FPP content in the full text. Relevant FPP content data can be supplemented to provide direct evidence. If this is difficult, please revise the relevant conclusion to a more rigorous one.

The text has been revised to address this point. Specifically, lines 138-140, page 4 now read:

“These results may be due to EpSS-L1 competing with the endogenous *N. benthamiana* squalene synthase for FPP precursor but this would need to be confirmed by assaying FPP levels.”

Page 5

In the first part of the results and discussion, there is a lack of detailed verification of key experimental steps. For example, when determining the functions of EpSS-L1 and EpSS-L2, only inference is made through expression in *Nicotiana benthamiana* and *Saccharomyces cerevisiae*, and there is no further direct evidence to confirm their mechanism of action.

We have now performed *in-vitro* assays on the *E. peplus* squalene and peplusol synthases and the hepta-swap mutants as detailed on pages 9 – 10 of the revised manuscript with results in Table 1, Fig. S7 and S8. These largely confirm the *in vivo* results, address the issue of endogenous squalene being produced in the host organisms that was raised by reviewer 3 and confirm the necessity for FPP for production of both squalene and peplusol.

Page 6 Line 187-190

“Given the high sequence homology with EpSSL-2 which predominantly exhibits squalene synthase activity (Fig 3) we suggest that EISS-L2 likely also encodes a squalene synthase.”

ELSS-L2 did not increase squalene production compared to the control. How can it be defined as a squalene synthase?

The reviewer makes a fair point if only squalene levels are considered. However, we have now also determined that peak 2 in the initial submission is 2,3-Oxidosqualene, a product of squalene oxidation. 2,3-Oxidosqualene is significantly increased in the *N. benthamiana* leaves transiently expressing *EISS-L2* (Fig 4C). We consider this as sufficient supporting evidence to propose that *EISSL-2* likely encodes squalene synthase.

We have revised the corresponding text as follows: Lines 215 - 218, pages 6:

“*EISS-L2* overexpression did not yield any peplusol, however it significantly increased the level of 2,3-oxidosqualene when expressed with and without *AtHMGRt* (Fig. 4C). Given the above result and high sequence homology with *EpSSL-2* which predominantly exhibits squalene synthase activity (Fig. 3) we suggest that *EISS-L2* likely also encodes a squalene synthase”.

Page 6

The study is mainly focused on a few Euphorbia species, which may not be representative of the entire genus. A larger number of species could be included to strengthen the generality of the findings.

In the introduction section of the manuscript we report the fact that peplusol production has been reported in four members of the *Euphorbia* genus and cite the relevant literature (lines 59-61, page 2). We focussed our study on the two species that we have access to. We consider that this introductory text clearly indicates that peplusol production occurs more widely in the *Euphorbia* genus.

Page 6-8

The manuscript infers the evolutionary origin of peplusol synthase relying on phylogenetic gene tree analysis and active site transplantation experiments. It will be better to combine other evolutionary analysis methods for more comprehensive understanding of the evolutionary relationship between peplusol synthase and squalene synthase?

We consider that the comprehensive phylogenetic gene tree analysis along with the experimental demonstration using the hexaswap and heptaswap approaches provide strong evidence that peplusol synthase has evolved from squalene synthase.

Minor comments:

Page 3 Line 83-91

When discussing the evolution of squalene synthase genes, it will be better to combine the discovery of peplusol synthase more closely?

We do not consider it would be appropriate to combine the discovery of peplusol synthase with squalene synthase in the introduction section of the manuscript since the discovery of peplusol synthase appears in the results and discussion. We consider the last two lines of the introduction, referring as they do to the discovery, functional

characterisation and evolutionary relationship of peplusol synthase with squalene synthase are sufficient for the introductory text.

Page 5 Line 155

The Squalene Synthase from *A. thaliana* has been defined as AtSS earlier. Thus, the abbreviation can be used in the subsequent text. Similar issues throughout the full text should also be corrected.

Thanks to the reviewer for pointing out this inconsistency. The text has been corrected throughout to be consistent with this convention.